# Diverse array of neutralizing antibodies elicited upon Spike Ferritin Nanoparticle vaccination in rhesus macaques

The repeat emergence of SARS-CoV-2 variants of concern (VoC) with decreased susceptibility to vaccine-elicited antibodies highlights the need to develop next-generation vaccine candidates that confer broad protection. Here we describe the antibody response induced by the SARS-CoV-2 Spike Ferritin Nanoparticle (SpFN) vaccine candidate adjuvanted with the Army Liposomal Formulation including QS21 (ALFQ) in non-human primates. By isolating and characterizing several monoclonal antibodies directed against the Spike Receptor Binding Domain (RBD), N-Terminal Domain (NTD), or the S2 Domain, we define the molecular recognition of vaccine-elicited cross-reactive monoclonal antibodies (mAbs) elicited by SpFN. We identify six neutralizing antibodies with broad sarbecovirus cross-reactivity that recapitulate serum polyclonal antibody responses. In particular, RBD mAb WRAIR-5001 binds to the conserved cryptic region with high affinity to sarbecovirus clades 1 and 2, including Omicron variants, while mAb WRAIR-5021 offers complete protection from B.1.617.2 (Delta) in a murine challenge study. Our data further highlight the ability of SpFN vaccination to stimulate cross-reactive B cells targeting conserved regions of the Spike with activity against SARS CoV-1 and SARS-CoV-2 variants.

SARS-CoV-2 infections continue to cause significant morbidity and mortality worldwide[1]. While vaccination is a fundamental tool to prevent SARS-CoV-2 infections and limit the COVID-19 pandemic, the continuous emergence of SARS-CoV-2 variants with increased neutralization resistance has raised serious concerns about the efficacy of the first-generation of vaccines and antibody therapeutics[2–4]. Viral variants, or variants of concern (VoC) such as Alpha (B.1.1.7), Beta (B.1.351), Gamma (P.1), Delta (B.1.617.2), and Omicron (B.1.1.529, BA.2, BA.5, BQ.1.1, XBB.1.5) subvariants, are characterized by increased infectivity, pathogenicity, or immune escape[5–9]. Specific population groups, such as age >60, immunocompromised individuals, or individuals with certain pre-existing conditions are particularly prone to breakthrough infections that can develop into life-threatening disease[10–12]. Emergence of the highly mutated SARS-CoV-2 VoC Omicron, with even greater escape capability, has further advanced vaccine formulations towards bivalent versions, or towards annually updated vaccines, that elicit immune responses against the most-recent VoC[1,13–18]. Previous studies have demonstrated that neutralization antibody titers against Omicron subvariants were low or undetectable after two immunizations of the monovalent WA-1-based Pfizer/BioNTech COVID-19 vaccine, while additional immunizations significantly boosted neutralizing antibodies against most Omicron subvariants[19,20]. However, even with subsequent boosting, breakthrough infections continue to occur[21,22], reinforcing the ongoing need for development of next-generation vaccines.

Previously we have reported the design and development of a SARS-CoV-2 ferritin nanoparticle-based vaccine candidate, SpFN (for Spike Ferritin Nanoparticle) administered with the Army Liposomal Formulation containing QS21 (ALFQ) adjuvant, which induced robust and broad immune responses in mice, Syrian golden hamsters, cynomolgus macaques, and rhesus macaques, resulting in protection against viral challenge[23–27]. SpFN is currently being evaluated in a

e-mail: skrebs@hivresearch.org; gjoyce@eidresearch.org

Phase I clinical trial (NCT04784767) for safety and immunogenicity. The SpFN molecule displays eight SARS-CoV-2 WA-1 Spike trimers on a self-assembling ferritin nanoparticle backbone and is administered together with the ALFQ adjuvant, a cholesterol-dense, liposomal composition that includes QS21 saponin[23]. Serum samples from SpFN-immunized rhesus macaques potently neutralized several SARS-CoV-2 VoC including the highly transmissible and pathogenic Delta and Omicron variants[23–25]. Neutralizing antibody activity after two doses of SpFN was an order of magnitude higher than that of convalescent serum samples[23]. In addition, potent neutralization titers were observed for another sarbecovirus, SARS-CoV-1, in both authentic and pseudotyped viruses[23,25]. The neutralization potency elicited by SpFN can likely be attributed to multiple aspects, including the repetitive array of the viral SARS-CoV-2 Spike glycoprotein on the ferritin nanoparticle[28–32], underlining the importance of multiplicity of antigen display to the immune system to generate robust and highly cross-reactive immune responses[24]. However, the targets of cross-neutralizing antibodies on the SARS-CoV-2 Spike after SpFN vaccination are currently unknown. Determination of the targets of SARS-CoV-2 neutralizing antibodies induced by novel vaccine strategies is key to understanding their mechanism of protection.

The aim of the current study was to define the epitopes targeted by antibodies at the molecular level induced by SpFN vaccination in non-human primates. To achieve this aim, we utilized a novel sorting strategy using both SARS-CoV-2 SpFN, and ferritin nanoparticles displaying SARS-1 Spike from the Urbani strain (SpFN$^1$), as probes to sort SpFN-specific B cells. Using this approach, we isolated mAbs targeting the N-Terminal Domain (NTD), Receptor Binding Domain (RBD), and S2 domain, where neutralizing antibodies were found to only target the NTD supersite[33,34] or RBD. Using alanine mutagenesis and hACE2 competition, we found that the RBD mAbs largely fell into two competition groups, one group that blocked hACE2 binding (Group A), and another that minimally inhibited hACE2 binding but instead targeted a cryptic, but conserved, epitope spanning across class III and V epitopes (Group B). We determined the X-ray crystal structures of the SARS-CoV-2 RBD in complex with representative neutralizing mAbs from these groups, termed the Walter Reed Army Institute of Research (WRAIR)−5021 and WRAIR-5001, respectively. These antibodies were tested for in vivo protection, structural analysis and binding and neutralization across diverse sarbecovirus and VoC to further elucidate the molecular mechanism of broad antibody recognition following SpFN vaccination.

## Results

### Isolation of SARS-CoV-2 neutralizing antibodies elicited in SpFN-vaccinated rhesus macaques

As previously reported, rhesus macaques were vaccinated twice, 4 weeks apart, with 50 μg of SpFN adjuvanted with the adjuvant Army Liposomal Formulation containing saponin QS-21 (ALFQ)[25] (Fig. 1a). Serum from the SpFN-vaccinated macaques were tested for neutralization using SARS-CoV-2 WA-1 pseudotyped virus and authentic virus assays, and epitope mapping using surface plasmon resonance (SPR). The magnitude of serum binding antibodies from SpFN-vaccinated macaques 2 weeks after the second vaccination were comparable to the magnitude of serum binding antibodies elicited by macaques 2 weeks after vaccination with two doses of mRNA-1273 (Supplementary Fig. 1a)[35,36]. Likewise, neutralizing antibody titers were previously published for both mRNA-1273 and SpFN-vaccinated rhesus macaques, and were found to be comparable at the 2 weeks post boost time point for both vaccine strategies[23,36–38]. Using a panel of SARS-CoV-2 mAbs to compete with plasma antibody binding to the Spike trimer[33,34,39–41], we were able to show distinct differences in the mapping of the polyclonal antibodies when comparing these two vaccination regimens (Supplementary Fig. 1b). Broad coverage of Spike antigenic sites for both vaccine regimens was observed; however

SpFN-derived plasma showed strong binding to RBD antigen site G, indicating a site targeted by class-III-like antibodies, while binding to this site was weak in mRNA-1273-derived plasma[33,39,42] (Supplementary Fig. 1b). Plasma from the SpFN-vaccinated macaques were tested for neutralization using SARS-CoV-2 WA-1 pseudotyped virus and authentic virus assays. Peak neutralization of SARS-CoV-2 was observed 2 weeks following the second dose (week 6) (Supplementary Fig. 1c), with cross-neutralizing activity against SARS-CoV-2 VoC and SARS-CoV-1 in authentic virus assays (Supplementary Fig. 1d). Robust levels of plasma IgG binding were detected to both SARS-CoV-1 S1 and RBD, and SARS-CoV-2 S1, HexaPro, RBD and NTD at the peak time point (week 6) and the necropsy timepoint week 9/10 (Supplementary Fig. 1e, f), whereas minimal IgM binding was detected across Spike domains (Supplementary Fig. 1g, h).

To understand the molecular targeting of broadly reactive antibodies elicited by SpFN vaccination, we isolated monoclonal antibodies from SpFN-vaccinated rhesus macaques, two weeks following the second SpFN immunization (Fig. 1a). Non-human primate (NHP).02 (animal Hs1606375, as previously published[23]) was selected based upon representative binding and neutralization profiles as shown in Supplementary Fig. 1. B cells from NHP.02 from the peak neutralization time point (week 6) were single-cell sorted using a novel, sequential sorting strategy that prioritized B cells binding to the vaccine SpFN molecules themselves to capture SARS-CoV-specific B cells (Supplementary Fig. 2). SpFN molecules containing either the Spike protein from SARS-CoV-2 WA-1 (SpFN) or SARS-CoV-1 Urbani (SpFN$^1$), in addition to fluorochrome-conjugated streptavidin tetramerized subdomains of the S protein (NTD and RBD), were used as probes to identify SARS-CoV-2- and/or SARS-CoV-1-specific B cells from animal PBMCs (Fig. 1b, Supplementary Fig. 2a, b). SpFN was used to mimic the SARS-CoV-2 virus with the goal of isolating mAbs targeting potential conformational or quaternary epitopes. This technique was previously used by our group to successfully bait both RBD- and NTD-directed B Cell Receptors (BCRs) from convalescent donors that potently neutralized SARS-CoV-2 when expressed as mAbs[34]. As a control, B cells from a naïve animal showed minimal reactivity to either SpFN molecules or tetramerized S protein subdomain baits (Supplementary Fig. 2c). The highest percentage of SARS-CoV antigen-positive B cells reacted to either one or both SpFN molecules, leaving only a minority co-staining to the tetramerized S protein subdomains RBD and NTD (Supplementary Fig. 2c, d).

In aggregate, 85 matched antibody heavy and light chain pairs were recovered and sequenced from single-cell SARS-CoV-specific B cells (Table S1). Of these, a total of 25 mAbs were subsequently produced as rhesus IgG1 in Expi293T cells based on complete variable region sequences after Sanger sequencing. Monoclonal antibodies were then screened for binding to SpFN and SpFN$^1$ molecules in an enzyme immunoassay (EIA) (Supplementary Fig. 3a, b). Based on reactivity to the SpFN and SpFN$^1$ molecules, 20 mAbs were further characterized in additional antibody binding assays. Four of the isolated mAbs, Walter Reed Army Institute of Research (WRAIR)-5002, -5008, -5014, and -5023 were found to bind to the empty ferritin particle (Fig. 1c, Supplementary Fig. 3c). However, this reactivity was limited to ferritin derived from *H. pylori*, with no cross-reactivity to ferritin derived from humans (Supplementary Fig. 3d, e).

The remaining 16 mAbs were tested for binding to a panel of 28 antigens spanning SARS-CoV-2, SARS-CoV-1, MERS-CoV, and the 4 seasonal coronaviruses 229E, HKU1, NL63, and OC43, using a multiplex bead-based Luminex assay (Supplementary Fig. 4a–c). Eleven of the mAbs were directed towards either the RBD or NTD, and three reacted to S2 (Fig. 1c, d). While most mAbs directed towards the NTD or RBD could also bind stabilized S protein (HexaPro), two of the mAbs were S protein-specific (did not bind NTD or RBD alone), and likely only recognize quaternary epitopes found on the stabilized trimer (Fig. 1c, d). More than half of the purified mAbs also cross-bound to the

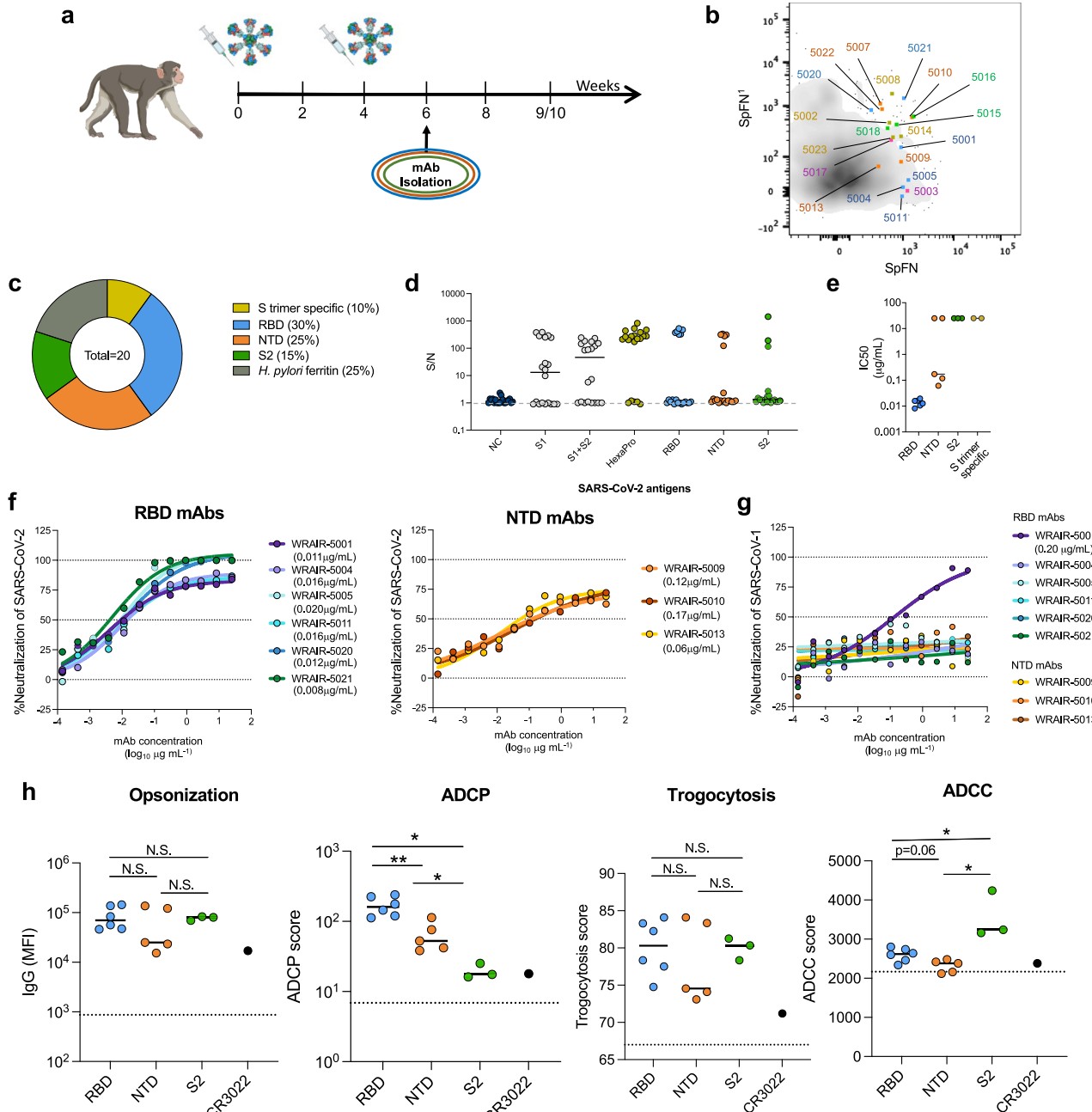

**Fig. 1 | Isolation of SARS-CoV-2 neutralizing antibodies elicited by SpFN-vaccinated rhesus macaques. a** SpFN vaccination timeline. Animals received a 50 µg dose at week 0 and 4. PBMCs from week 6 (2 weeks post-boost) were used for B cell sorting and mAb isolation. **b** Reactivity of isolated WRAIR mAbs towards SpFN[1] (expressing SARS-CoV-1 S protein) and SpFN (expressing SARS-CoV-2 S protein) through flow cytometry as measured during sorting. **c** The proportion of isolated mAbs binding to *H. pylori* ferritin, subdomains of S protein or the stabilized S protein (HexaPro). **d** Binding to SARS-CoV-2 antigens or the stabilized S protein (HexaPro) in a multiplexed bead-based assay by twenty purified monoclonal antibodies. **e** Neutralization potency of isolated WRAIR mAbs segregated by S protein subdomain binding specificity. Shown are the $IC_{50}$ values (µg ml$^{-1}$) from the SARS-CoV-2 (IL1/2020) pseudotyped assay, calculated from three independent experiments. **f** Neutralization curves of the (left) RBD-directed and (right) NTD-directed neutralizing mAbs against SARS-CoV-2 (WA-1). Shown in parentheses are the $IC_{50}$

values for each mAb. **g** Neutralization of SARS-CoV-1 as measured in pseudotyped viral inhibition assays, with the $IC_{50}$ value for WRAIR-5001 shown in parentheses. Plotted are the mean ± s.e.m. from two independent experiments. **h** Assessment of RBD (*n* = 6), NTD (*n* = 5) and S2 (*n* = 3) -directed mAbs in Fc-mediated effector functions including antibody-dependent cell surface S binding (opsonization), cellular phagocytosis (ADCP) (from top to bottom: *$P$ = 0.02, **$P$ = 0.009, *$P$ = 0.04), cell membrane transfer (trogocytosis), and cellular cytotoxicity (ADCC) (from top to bottom: *$P$ = 0.02, *$P$ = 0.04). Black horizontal lines indicate the mean value and asterisks represent significance by two-tailed Mann–Whitney *t*-test. The dotted line indicates the positivity threshold as determined by non-SARS-CoV-2 (negative) monoclonal antibody control (Zika mAb rhMZ134). Monoclonal antibody CR3022 is used as a positive control. Each data point is the mean of duplicate data from a single experiment. Source data are provided as a Source Data file. The rhesus macaque image was created with BioRender.com.

same domains of SARS-CoV-1 (Supplementary Fig 4a–c). Potent neutralization activity, as measured with pseudotyped virus (WA-1), was observed only for RBD and NTD mAbs, ranging from subnanomolar to micromolar concentrations (Fig. 1e, f). All the isolated RBD-specific mAbs neutralized, while 60% of NTD-specific mAbs, and none of the S2- or HexaPro-specific mAbs harbored neutralization activity against SARS-CoV-2 WA-1 (Fig. 1e, f, Supplementary Fig. 4d). Neutralization plateaued around 75% for WRAIR mAbs targeting NTD (Fig. 1f), as previously described[34]. One RBD-directed mAb, WRAIR-5001, demonstrated neutralization activity against pseudotyped SARS-CoV-1 at 0.20 µg/ml (Fig. 1g).

As a functional Fc region may enhance the protective efficacy of neutralizing antibodies given therapeutically in SARS-CoV-2 mouse models[43] we tested the ability of the purified mAbs to facilitate Fc effector functions, in vitro. The ability to facilitate cell surface S protein binding (opsonization), a function wherein either neutralizing or non-neutralizing antibodies surround viral or cellular components and often associated with enhanced viral clearance through phagocytosis, did not vary based on epitope targeted (Fig. 1h). Opsonization was previously reported by our group to be higher in NTD-directed mAbs isolated from SARS-CoV-2 convalescent donors[34]. Compared to S2-directed mAbs, RBD- and NTD-directed mAbs mediated higher levels of antibody-dependent cellular phagocytosis (ADCP), with RBD-directed mAbs being superior to NTD-directed mAbs (Fig. 1h). ADCP is a method of viral clearance mediated through the antibody Fc domain commonly associated with monocytes[44–46] (Fig. 1h). Interestingly, these results differed from our previous observations with mAbs isolated from WA-1 convalescent donors, wherein NTD-directed mAbs performed ADCP at higher levels compared to RBD-directed mAbs[34]. In contrast, S2-directed mAbs had higher functionality compared to the RBD and NTD-directed mAbs in antibody-dependent cellular cytotoxicity assays (ADCC) (Fig. 1h), a function also found at high levels in the serum of SARS-CoV-2 convalescent donors and thought to be most frequently facilitated by natural killer cells[47–50]. Together, these data indicate that two doses of SpFN induced antibodies that targeted a wide range of neutralizing epitopes with cross-sarbecovirus neutralizing activity, while also facilitating Fc effector functions associated with therapeutic and natural protection.

**SpFN vaccination elicits antibodies directed toward multiple epitopes.** Next, we sought to define the epitope specificity of the isolated mAbs by measuring the percent residual binding to either the Spike NTD or RBD, in the presence of control mAbs with a previously defined epitope. Using a set of previously defined competing WRAIR mAbs[34], the NTD-directed mAbs fell into three competition groups (Fig. 2a). Of the NTD-directed mAbs, Group B mAbs (WRAIR-5009, -5010, and -5013) were the only mAbs that neutralized WA-1, while showing no cross-neutralization to other SARS-CoV-2 VoC (Fig. 1f and Supplementary Fig. 4e). These Group B NTD mAbs were capable of cross-binding SARS-CoV-1, but were unable to neutralize SARS-CoV-1 (Supplementary Fig. 4a, e). We compared these epitopes to that of the polyclonal serum (Supplementary Fig. 1), where SpFN-derived serum antibodies blocked binding to NTD antigen site corresponding to control mAb S652-118. Similarly, Group B mAbs WRAIR-5009, -5010, and -5013 also competed for binding to this epitope (Supplemental Fig. 5a) The epitopes of all NTD-targeted mAbs appear to lie near or overlap with the NTD supersite (Fig. 2a, inset). Most mutations found in the VoC are found on the NTD face distal to the rest of the Spike, explaining the limited cross-neutralization activity of these mAbs against VoC.

RBD directed mAbs fell into previously described WRAIR RBD competition Groups A and B (Fig. 2b). The Group A RBD mAbs (WRAIR-5005, -5020, and -5021) facilitated complete neutralization of SARS-CoV-2 WA-1, while Group B RBD mAbs (WRAIR-5001, -5004, -5011) were incapable of mediating complete viral neutralization of SARS-CoV-2

WA-1, peaking around 75% neutralization, using pseudotyped virus (Fig. 1f). Group B RBD mAbs were able to bind SARS-CoV-1 RBD (Supplementary Fig 4b), with WRAIR-5001 demonstrating neutralization activity against SARS-CoV-1 (Fig. 1g). Shotgun alanine mutagenesis epitope mapping revealed that RBD residues critical for mAb interaction were conserved within Group A and overlapped with previously identified class I RBD epitopes[40] (Fig. 2c, d). Residues critical for mAb interaction were also conserved within Group B, and overlapped with epitopes spanning class III and V RBD epitopes that were previously identified[40] (Fig. 2c, d). We compared these epitopes to that of the polyclonal serum (Supplementary Fig. 1b), where we used the same panel of SARS-CoV-2 mAbs that were used to map the specificity of the serum polyclonal responses to compete with these WRAIR mAbs (Supplementary Fig. 6a). SpFN vaccination elicited polyclonal serum responses targeting many known protective RBD epitopes (Supplementary Fig. 1b), and uniquely mounted a polyclonal response corresponding to a neutralizing class III epitope (Fig. 2b). Similarly, WRAIR Group A and B mAbs overlapped with epitopes targeted by class I/II and class III/V mAbs, including the unique class III epitope targeted by S309 (Supplemental Fig. 6a, b). Using viral escape assays, we determined residues that contributed to escape from two representative Group A RBD mAbs (WRAIR-5001, and -5011) and one mAb from Group B RBD (WRAIR-5021) (Fig. 2d, e), which confirmed the epitope targeting of the mAbs. Combined, these data map the antibody specificity following SpFN vaccination and demonstrate that cross-neutralizing epitopes are located within the RBD.

**In vivo efficacy of mAbs elicited after SpFN vaccination against VoC Delta.** We next sought to further explore the ACE2 inhibition and breadth of neutralization potency against SARS-CoV-2 VoC and determine if these mAbs could protect against a SARS-CoV-2 VoC viral challenge. None of the NTD- or S2-directed mAbs were capable of blocking hACE2 binding (Supplementary Fig. 5b, c). Group A RBD mAbs (WRAIR-5005, -5020, -5021) completely blocked hACE2 binding to both RBD and the stabilized Spike trimer S-2P, whereas Group B RBD mAbs (WRAIR-5001, -5004, -5011) had incomplete hACE2 blocking activity against both RBD and the Spike trimer S-2P (Fig. 3a, b). Group A RBD mAbs demonstrated binding across early-pandemic VoC RBD molecules (Fig. 3c). However, interaction with Omicron VoC or SARS-CoV-1 RBD molecules was severely impacted or completely abrogated (Fig. 3c). Similarly, potent neutralization by the Group A RBD WRAIR mAbs was observed across WA-1, Beta (B.1.351) and Delta (B.1.617.2) VoCs, but a lack of neutralization was found to Omicron subvariants and SARS-CoV-1 (Fig. 3d, Table S2). Conversely, WRAIR Group B RBD mAbs demonstrated significant binding breadth across all SARS-CoV-2 VoC RBDs, and none of the Group B RBD mAbs were affected by any of the variant mutations, including Omicron subvariant mutations (Fig. 3c). Modest neutralization was observed across Beta (B.1.351), Delta (B.1.617.2), and Omicron (BA.1, BA.2), but a lack of neutralization was found to Omicron BA.4/5 (Fig. 3d, Table S2). WRAIR-5001 displayed both cross-binding activity to SARS-CoV-1 RBD and neutralization to SARS-CoV-1 (Figs. 1g and 3c, d), and was further evaluated for affinity to SARS-CoV-2 VoC and SARS-CoV-1 RBD molecules. Interestingly, while WRAIR-5001 was able to bind with high affinity to SARS-CoV-2 Omicron BA.1 and BA.4/5 ($K_D = 0.23$ nM and 0.92 nM, respectively Fig. 2g), this mAb had diminished neutralization activity to these subvariants (Fig. 3d, e). In contrast, for SARS-COV-1, WRAIR-5001 displayed modest affinity ($K_D = 3.45$ nM), with strong neutralization ($IC_{50} = 0.22$ µg/ml, Fig. 3d, e).

We showed in previous studies that SpFN-vaccinated rhesus macaques were protected against viral challenge[25]. To determine if protection from challenge was due to targeting the defined epitopes of our isolated mAbs, one representative mAb from each RBD competition group (Group A, WRAIR-5021 and Group B, WRAIR-5001) were further tested for in vivo protection against lethal SARS-CoV-2

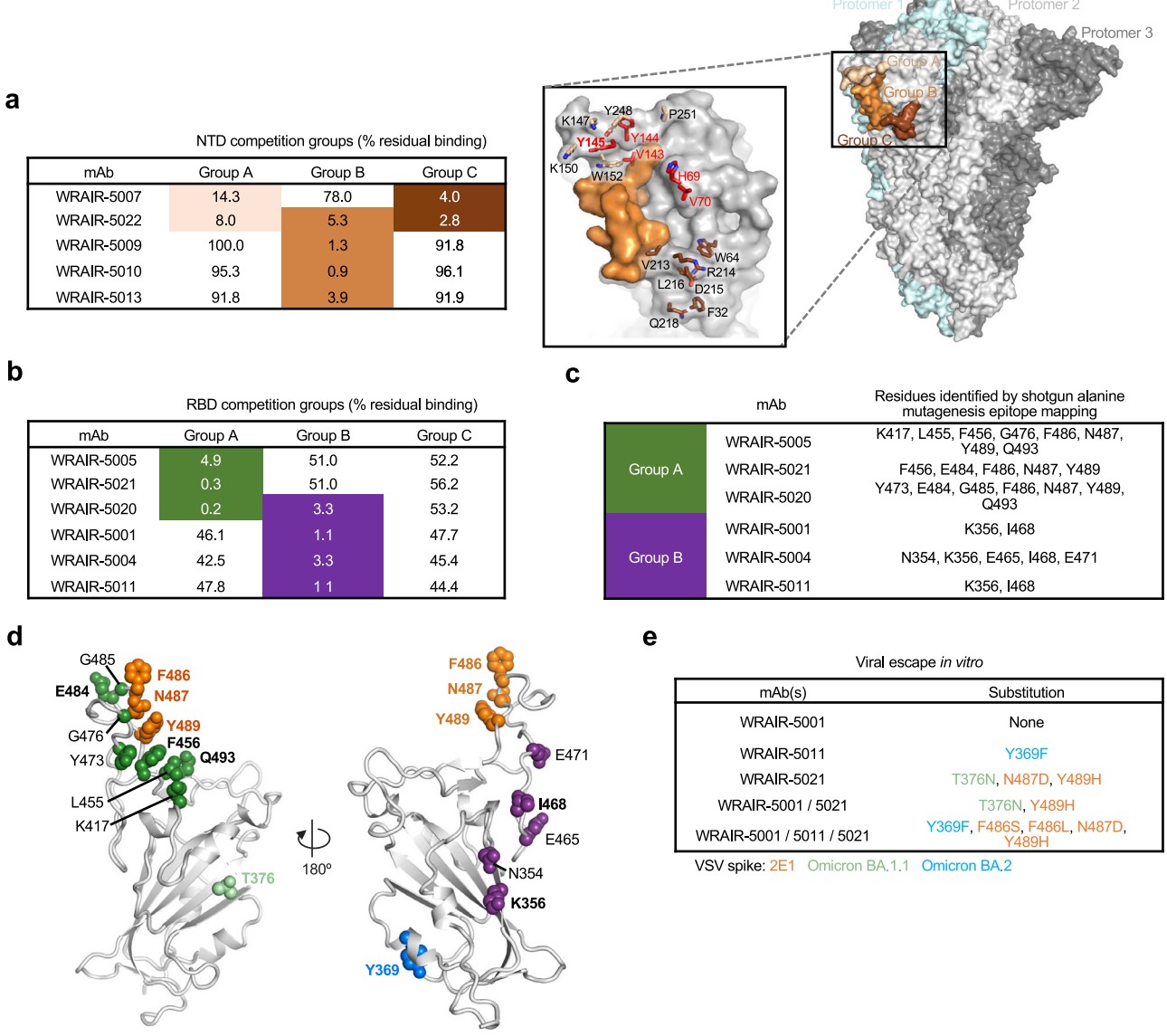

**Fig. 2 | Epitope binning of WRAIR NTD and RBD-targeted mAbs against variants of concern.** Epitope binning of **a** NTD-directed or **b** RBD-directed mAbs as measured through a BLI-based competition assay. Values are the percentage of residual binding of the indicated WRAIR second antibody after saturation of the antigen (NTD or RBD subdomain) with a representative first antibody (NTD: Group A: WRAIR-2025, Group B: WRAIR-2137, Group C: WRAIR-2054) (RBD: Group A: WRAIR-2125, Group B: WRAIR-2063, Group C: WRAIR-2151). Competition groups are indicated by boxes in shades of brown (NTD) or green/purple (RBD). **(Right)** Closed S trimer (PDB 6ZGE) with epitopes of the NTD-targeted mAb competition groups indicated in tan, light brown, and dark brown. Each protomer of the S trimer is colored cyan, light grey, or dark grey. (Inset **a**, center) The NTD is shown in surface representation, with residues identified for competition shown in respective shades of brown. Residue deletions observed in Omicron subvariants are highlighted in red. **c** Mapping of RBD A and B mAbs using alanine mutagenesis across the spike glycoprotein. **d** Residues identified by **e** viral escape assays are highlighted to show the targeted epitope of Group A and Group B RBD WRAIR mAbs. Residues of each group are shown in sphere representation on the SARS-CoV-2 RBD. Residues targeted by more than one antibody are highlighted. Source data are provided as a Source Data file.

challenge with the circulating VoC at the time of experimentation, Delta (B.1.617.2), in the K18-hACE2 transgenic mouse model. We anticipated that both mAbs would confer 100% protection against a lethal WA-1 viral challenge, based on their neutralization titers to WA-1 correlating with in vivo protection for RBD mAbs[34]. Against Delta, WRAIR-5021 demonstrated potent neutralization (IC$_{50}$ = 0.002 μg/ml), whereas WRAIR-5001 yielded more modest neutralization (IC$_{50}$ = 1.6 μg/ml, Fig. 3d). Prophylactic in vivo protection was assessed by intravenous administration of WRAIR-5001 or WRAIR-5021 at a dose equivalent to 10 mg/kg, side-by-side with control mAb WRAIR-2125[34], (positive control SARS-CoV-2 neutralizing mAb isolated from a

convalescent donor), and an IgG isotype control (Zika mAb MZ4), 24 hours prior to administration of SARS-CoV-2 Delta (B.1.617.2) challenge virus (Fig. 3f).

At 48 hours post-challenge, 5 animals from each group were sacrificed for viral replication analysis. The Group A RBD WRAIR-5021 group showed reduced viral replication in the lung and bronchoalveolar lavage (BAL) of challenged mice at a level akin to the WRAIR-2125 positive control, and significantly better than the IgG isotype negative control ($P < 0.0001$) (Fig. 3g). However, there was no significant difference in the viral replication in the lung or BAL between the Group B WRAIR-5001 or the IgG isotype negative control group

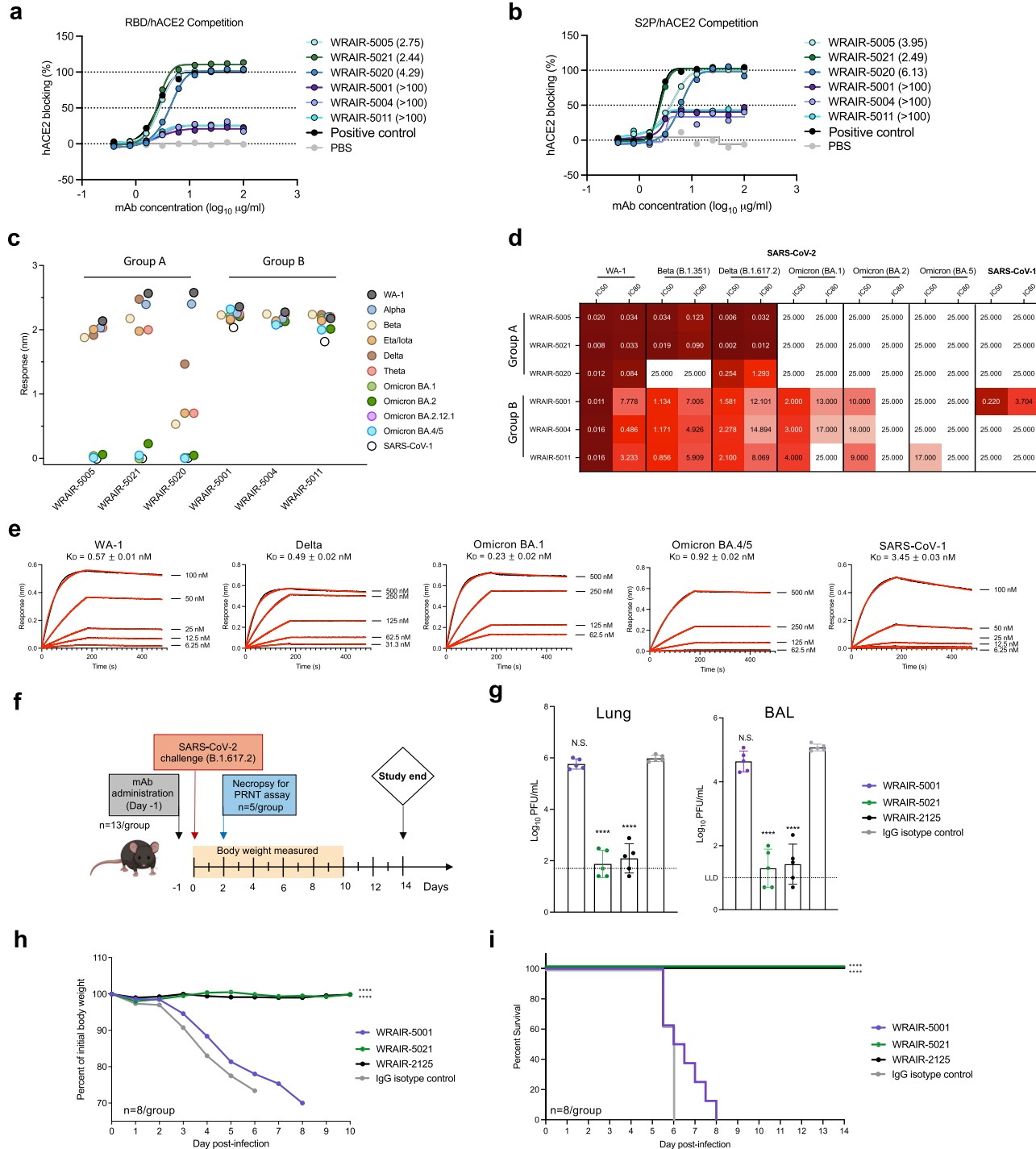

**Fig. 3 | Select RBD mAbs demonstrate ACE2 binding inhibition, neutralize across SARS-CoV-2 variants of concern, and demonstrate protection in vivo.** ACE2 inhibition by WRAIR RBD mAbs in a BLI-based assay. RBD mAbs were assessed for their ability to block hACE2 binding to **a** SARS-CoV-2 RBD or **b** S-2P. The half-maximal effective concentration (EC$_{50}$) in µg ml$^{-1}$ is indicated in parentheses. Cross-reactivity of mAbs were assessed for activity against SARS-CoV-2 VoCs and SARS-CoV-1, using **c** BLI, or **d** IC$_{50}$ values measured against pseudotyped virus. **e** Binding kinetics of WRAIR-5001 with SARS-CoV-2 WA-1, Delta, Omicron BA.1, Omicron BA.4/5, and SARS-CoV-1 RBDs, as measured by BLI. Additional kinetics values may be found in Supplementary Table 2. **f** In vivo protection study plan and design of prophylactic administration of RBD mAbs. RBD mAbs WRAIR-5001 and WRAIR-5021 were given at single dose of 10 mg/kg intravenously in K18-hACE2 transgenic mice, followed by challenge with SARS-CoV-2 Delta (B.1.617.2) intranasally 24 hours

later (*n* = 13 animals/group). 5 mice from each group were sacrificed on day 2 for plaque reduction neutralization test (PRNT) assay, and surviving mice were measured for body weight changes and survival out to day 14. RBD mAb WRAIR-2125 was used a positive control, and the negative control was an IgG isotype Zika-specific mAb, MZ4. **g** Particle forming units (PFU) measured in the lungs or bronchoalveolar lavage (BAL) of mice on study day 2 (*n* = 5 animals/group). Error bars indicate the standard deviation. (****$P$ < 0.0001 using ordinary one-way ANOVA compared to IgG isotype control, MZ4) **h** Percent loss of body weight out to study day 10 and **i** survival curves out to study day 14 (study end date). (****$P$ < 0.0001 using ordinary one-way ANOVA compared to IgG isotype control, MZ4). Source data are provided as a Source Data file. The mouse image was created with BioRender.com.

(Fig. 3g). The remaining animals (n = 8) in each group were monitored for loss of body weight out to 10 days and survival out to 14 days. Group A RBD WRAIR-5021 conferred 100% in vivo protection against body weight loss and lethal challenge (P < 0.0001) (Fig. 3h, i). Group B RBD WRAIR-5001 was able to extend the median survival by two days compared to the IgG isotype-negative control, but was unable to completely protect against weight loss and death (Fig. 3h, i). The lack of protection by Group B WRAIR-5001 to Delta (B.1.617.2) aligned with modest neutralization of WRAIR-5001 to Delta (IC$_{50}$ = 1.581 µg/ml) in vitro, whereas Group A WRAIR-5021 was able to potently neutralize Delta, and completely inhibit hACE2 binding to both RBD and S-2P. (IC$_{50}$ = 0.002 µg/ml) (Fig. 3a, b, d).

**Crystal structure of Group A RBD mAb, WRAIR-5021, in complex with SARS-CoV-2 RBD.** To further define the molecular recognition of RBD mAbs elicited by SpFN vaccination that demonstrated VoC neutralization, we performed structural studies of Group A RBD mAb WRAIR-5021. Binding competition assay-based epitope mapping experiments, using previously reported antibodies, indicated the hACE2 binding site as the target for RBD-A antibodies (Fig. 3a, b). To understand the structural basis of RBD recognition by these mAbs, we crystallized Group A WRAIR-5021 Fab in complex with the SARS-CoV-2 RBD and analyzed the structure at a final resolution of 2.3 Å (Fig. 4a and Table S3). Structure determination confirmed that WRAIR-5021, which potently neutralizes SARS-CoV-2 (Fig. 1f) targets the hACE2 binding site with a distinct epitope centered on the hACE2 binding ridge loop (residues 475-478 and 484-490) (Fig. 4b). Structure superimposition of the WRAIR-5021-RBD complex structure onto the hACE2-bound RBD structure (PDB code: 6M0J) revealed nineteen of the twenty-five hACE2 binding residues are part of the WRAIR-5021 epitope, demonstrating 76% epitope-binding site overlap. WRAIR-5021 forms extensive interactions across the entire length of the receptor binding motif, with a BSA of 1221.6 Å$^2$ with heavy and light chains contributing 61.9% and 38.1% of the total BSA, respectively (Fig. 4c). Heavy and light chain interactions form a total of 13 and 9 hydrogen bonds, respectively, with the light chain forming three additional salt-bridge interactions. WRAIR-5021 heavy chain contacts are mediated by all the CDR loops with CDR H1 contributing the most, covering >300 Å$^2$ of the RBD interface (Fig. 4c). Major heavy chain contacts are formed by a set of Tyr residues in the CDR H1 and H2 (residues Tyr27, Tyr33, Tyr34, and Tyr51). Light chain contacts are primarily mediated by CDR L2 and L3, with limited contributions from CDR L1. Major light chain contacts are mediated by a set of polar and charged residues in the CDR L1-L3 (Ser30, Asp50, R53, Ser56, Asp92, Ser93and Asp94). (Fig. 4d and Table S4). WRAIR-5021 exhibited low levels of somatic hypermutation (SHM) with heavy and light chain V-genes consisting of 4 and 6 changes, respectively. Of these mutations, 2 residues of the heavy chain, Thr53 and Thr54, are found at the interface with the RBD, with the Thr54 hydroxyl group hydrogen bonded to the carboxyl side chain of RBD Glu484. From the light chain, the 2 nitrogens of the Arg53 guanidino group form hydrogen bonds with the carboxyl side chains of RBD residues Asp420 and Asn460, and Phe32 and Asp50 are ~54% and ~28% buried at the interface.

Next, we performed structural superimposition of WRAIR-5021 with representative antibodies from previously defined classes[41]. Based on this analysis, we confirmed that WRAIR-5021 can be classified as a class I mAb (Fig. 4e). With respect to VoC, except Delta variant, all the RBD mutations in the Alpha, Beta and Gamma VoC are within the epitope site of WRAIR-5021 mAb. However, based on the structure modeling, none of these mutations, including those found in the Delta variant, would offer enough steric/electrostatic clashing to prevent RBD binding by WRAIR-5021 (Fig. 4f, Supplementary Fig. 7a–d), suggesting a molecular mechanism for the in vivo protection against Delta challenge. Assessment of WRAIR-5021 binding by BLI, against a panel of RBD molecules showed high-affinity nanomolar binding to WA-1,

Delta, and Omicron subvariants BA.1 and BA.4/5 (Table S2). In the case of the Omicron BA.1 variant, 9 of the 15 RBD mutations overlap with the WRAIR-5021 epitope, with many of the mutations altering the epitope structure, and were expected to negatively impact Omicron VoC binding (Fig. 4f). However, we did not observe significant changes in affinity, indicating some plasticity within the epitope, like that seen for hACE2-binding.

To further understand the WRAIR-5021 epitope in the context of full-length Spike, we modeled WRAIR-5021 to the closed conformation (all RBD down; PDB code: 6VXX), and 1- or 2-RBD in the up conformations (PDB codes: 7DWZ and 6X2B) (Fig. 4g)[51–53]. Structure superimposition demonstrated that despite a few minor clashes, the binding of WRAIR-5021 is compatible with both the up and down conformations of S protein, indicating accessibility of the epitope on the Spike trimer. Antibodies similar to WRAIR-5021 may arise from the precursor germline VH4-99*01 in rhesus macaques with minimal SHM (Fig. 4h).

**Crystal structure of RBD-B mAb, WRAIR-5001, in complex with SARS-CoV-2 RBD.** Previously, we have described a set of RBD neutralizing antibodies (WRAIR-2057, WRAIR-2063, and WRAIR-2134) isolated from a convalescent human subject which were of particular interest because of their novel epitope on the SARS-CoV-2 RBD[34,54]. Our binding competition and alanine scanning experiments indicated that WRAIR-5001 also targets a similar epitope (Fig. 2c and Supplementary Fig. 6a). To further our knowledge of this epitope, we determined the crystal structure of the WRAIR-5001 Fab in complex with SARS-CoV-2 RBD. The crystal structure of the WRAIR-5001-RBD complex was determined to a final resolution of 4.3 Å and refined to an R$_{work}$/R$_{free}$ of ~0.25/0.30 (Table S3). WRAIR-5001 binds to a less typical, cryptic epitope, located on the "side" of the RBD, distal from the hACE2 binding site (Fig. 5a–c). Overall, the WRAIR-5001 epitope covers a total BSA of 1087.3 Å$^2$ with heavy and light chains contributing 26.6% and 73.4% of total BSA, respectively (Fig. 5d and Table S5). WRAIR-5001 recognition of SARS-CoV-2 RBD is primarily based on CDR H2, H3, and CDR L1-L3 loops. Heavy and light chain interactions form a total of 2 and 9 hydrogen bonds, respectively, with the light chain forming three additional salt-bridge interactions. WRAIR-5001 heavy chain contacts are mediated by CDR H2 and H3 with CDR H3 contributing the most, covering ~200 Å$^2$ of the RBD interface (Fig. 5d). Major heavy chain contacts are formed by a set of hydrophobic residues (Trp47, Trp50, Val57, Val59, Leu103, Val104 and Val105). Leu103 inserts into a hydrophobic pocket on the RBD (formed by Trp353, Arg355 and Phe464) and contributes 127.1 Å$^2$ of the binding interface area. Light chain contacts are primarily mediated by CDR L1 and L3, with limited contributions from CDR L2. Major CDR L1 contacts are mediated by a set of polar and charged residues (Asp25, Asn26, Ala28, Ser29, Lys30 and Asn31). Asn26 inserts in a shallow pocket lined by RBD residues Glu340-Thr345 and buries 73.3 Å$^2$ of the surface area (Fig. 5e and Tables S5). Major CDR L3 contacts are mediated by a set of hydrophobic residues (Trp90, Tyr92-93, and His96) with Tyr93 and His96 contributing the most. Tyr93 inserts into a deep pocket lined by RBD residues Val341, Ala344, Phe347 and Ser399, and buries 143.4 Å$^2$ of the surface area. His96 inserts into a deep hydrophobic pocket lined by RBD residues Ala351, Tyr352, Arg466 and Ile468, and buries 105.7 Å$^2$ of the surface area (Fig. 5e and Tables S5). WRAIR-5001 has low levels of somatic hypermutation (SHM) with heavy and light chain V-genes consisting of 7 and 6 changes, respectively. Of these SHMs, 2 residues of the heavy chain, Val56 and Val58, are ~46% and 67% buried at the RBD interface. In the light chain the carboxamide of Asn32 is hydrogen bonded to the backbone amine and carboxyl oxygen of RBD Arg357, while Asn66 and Ala29 are 34% and 100% buried, respectively, at the RBD interface (Supplementary Fig. 8).

Structural superimposition of WRAIR-5001 with representative antibodies from previously defined classes indicated that the

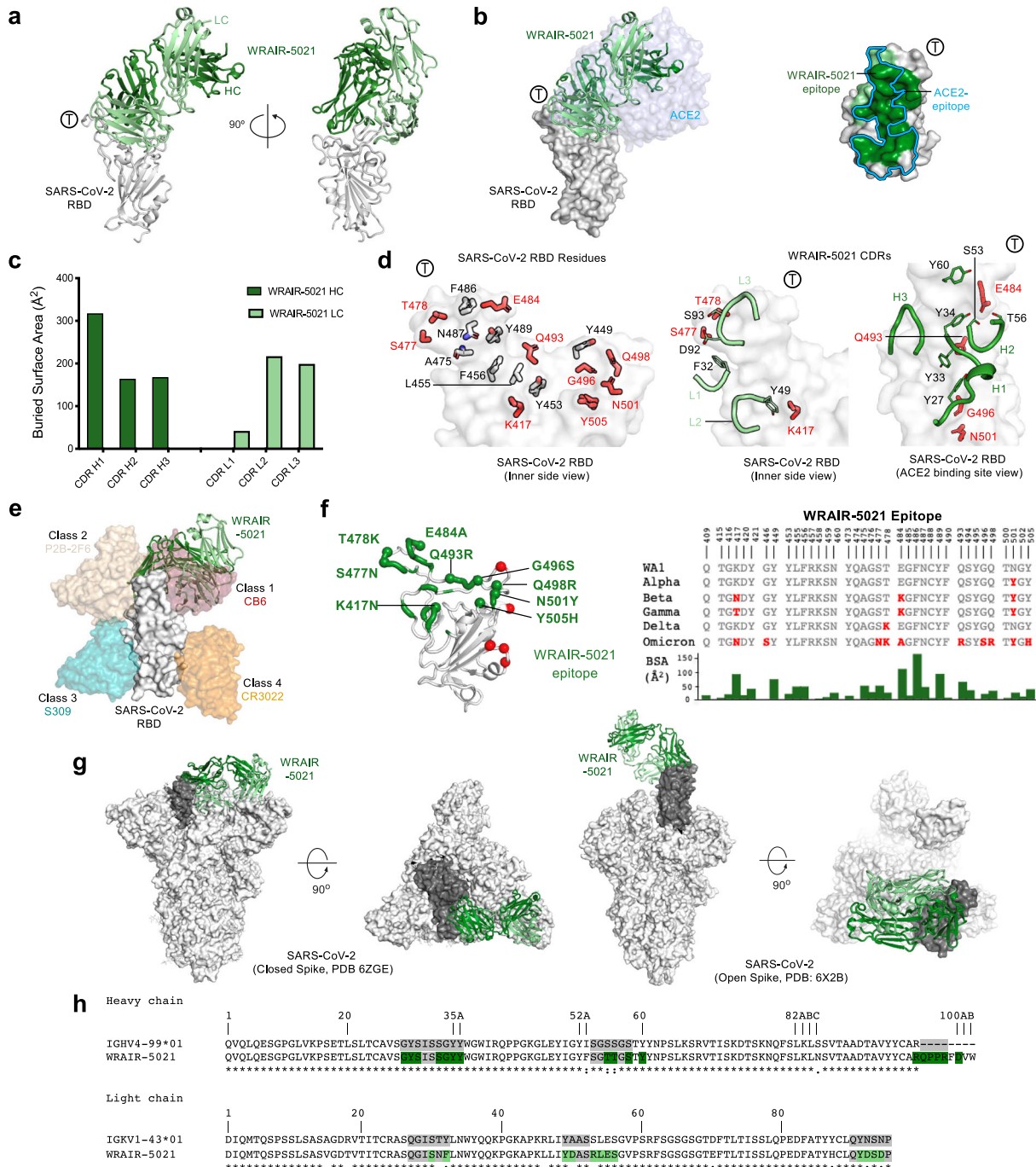

**Fig. 4 | Structure and epitope analysis of WRAIR-5021. a** (Left) Crystal structure of WRAIR-5021, in complex with SARS-CoV-2 RBD (white) shown in cartoon representation. WRAIR-5021 heavy and light chains are colored dark and light green, respectively (color scheme applies to all panels in Fig. 4). The ACE2 binding ridge is indicated by ⓣ. (Right) Structure is shown at 90° rotation. **b** (Left) Overlay of SARS-CoV-2 RBD bound ACE2 structure (PDB: 6M0J) onto the WRAIR-5021-RBD complex structure. ACE2 is shown in light blue/grey surface. (Right) Epitope of WRAIR-5021 shown on the surface of the RBD. The ACE2 epitope is outlined in cyan. **c** Buried surface area (BSA) for the CDR loops is shown as a bar diagram. **d** (Left) Key antibody contacting residues of RBD are shown as sticks, with residues reported in VoCs in red. (Right) Important heavy and light chain contacting residues shown as thin sticks. RBD residues reported in VoCs are represented in red sticks. **e** Structure of the WRAIR-5001-RBD complex overlaid onto previously reported antibodies in complex with SARS-CoV-2 RBD (representing frequently observed SARS-CoV-2

epitopes[39]). **f** (Left) Omicron mutations highlighted as red spheres on the surface of SARS-CoV-2 RBD. The WRAIR-5021 epitope is shown in tubular representation and colored dark green. Omicron mutations that fall within the mAb epitope are shown as green spheres and labeled. (Right) RBD sequence alignment with WRAIR-5021 epitope indicated. Mutated residues in VoCs are highlighted in red. BSA for epitope residues are shown in the bar graph at the bottom. **g** Structural superimposition of the WRAIR-5021-RBD complex with closed (all RBD down conformation, PDB code: 6ZGE) and open (1-RBD-up, PDB: 6X2B) conformations of SARS-CoV-2 S-2P. WRAIR-5021-RBD is overlaid onto the RBD (dark gray surface) from one protomer. Side and top views are shown. **h** Sequence alignment of WRAIR-5021 with its precursor germline gene. CDRs are shaded grey, with residue numbering and CDR loops designated using the Kabat system. Residues interacting with the RBD are colored green. Symbols *,:, and . denote identical, similar, and less similar residues, respectively. Source data are provided as a Source Data file.

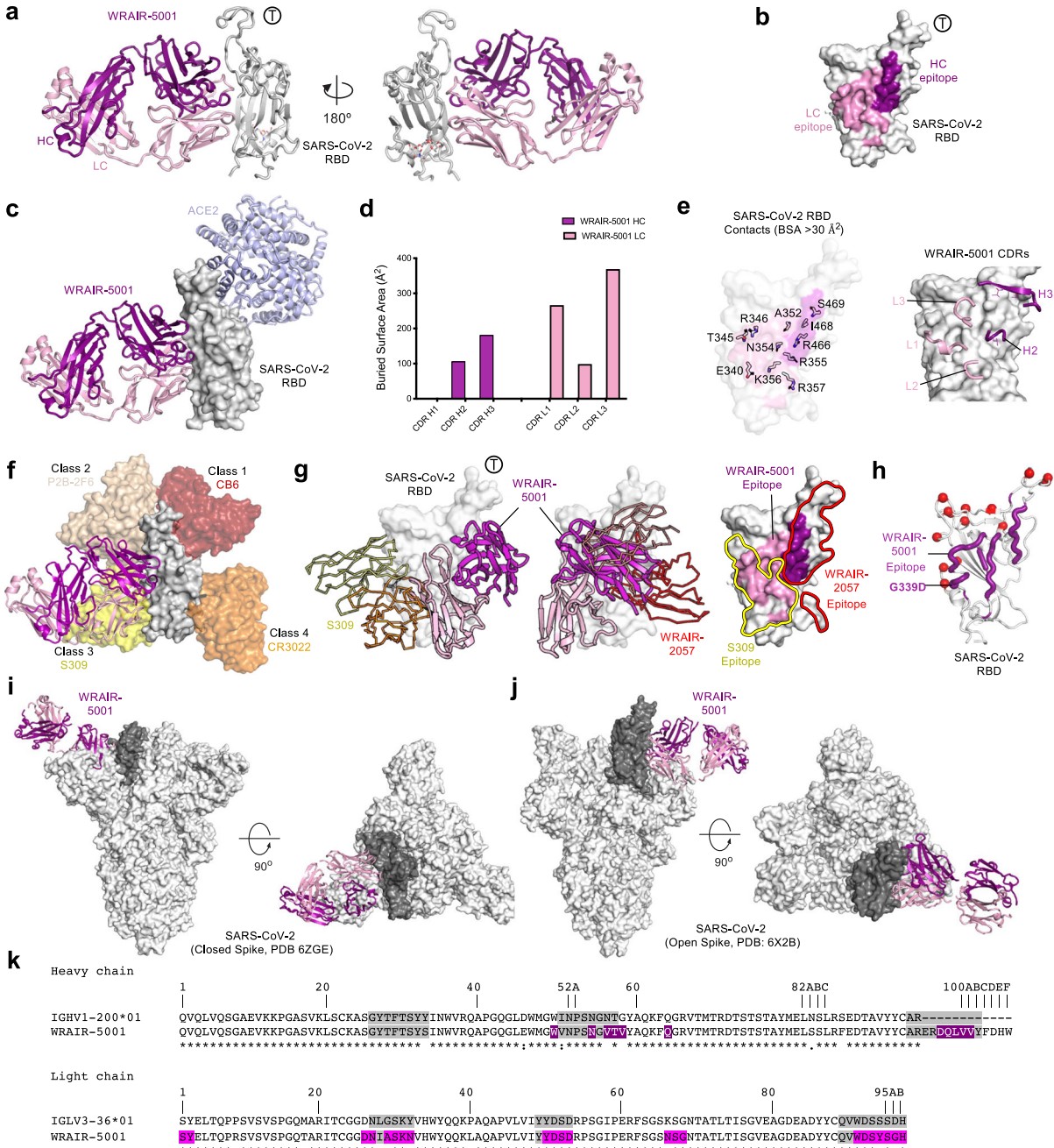

**Fig. 5 | Structure and epitope analysis of WRAIR-5001. a** (Left) Crystal structure of WRAIR-5001, in complex with SARS-CoV-2 RBD (white) shown in cartoon representation. Heavy and light chains of WRAIR-5001 are colored dark and light pink, respectively (color scheme applies to all panels in Fig. 5). The ACE2 binding ridge is indicated by ⓣ. (Right) Structure is shown at 180° rotation. **b** SARS-CoV-2 RBD shown in surface representation. **c** Overlay of the SARS-CoV-2 RBD bound ACE2 structure (PDB: 6M0J) onto the WRAIR-5001-RBD complex structure. ACE2 is shown in cartoon representation and colored light blue/grey. **d** Buried surface area (BSA) for the WRAIR-5001 heavy and light chain CDR loops shown as a bar diagram. **e** (Left) Key antibody contacting residues of RBD shown as sticks. (Right) Important heavy and light chain contacting residues of contributing CDRs shown as thin sticks and labeled as per antibody coloring scheme. **f** Structure of WRAIR-5001-RBD complex overlaid onto previously reported antibodies in complex with SARS-CoV-2 RBD (representing frequently observed SARS-CoV-2 epitopes)[39]. **g** (Left-Middle) Crystal structures of S309-RBD and WRAIR-2057-RBD complexes overlaid onto the WRAIR-5001-RBD structure. Antibodies S309 and WRAIR-2057 are shown in ribbon representation and colored yellow and red, respectively. (Right) The WRAIR-5001 epitope is colored light and dark pink on the surface of SARS-CoV-2 RBD while S309 and WRAIR-2057 epitopes are outlined and labeled accordingly. **h** Omicron mutations highlighted as red spheres on the surface of SARS-CoV-2 RBD. The WRAIR-5001 epitope is shown in tubular representation and colored dark pink. **i, j** Structural superimposition of the WRAIR-5001-RBD complex with closed (all RBD down conformation, PDB code: 6ZGE) and open (1-RBD-up, PDB: 6X2B) conformations of SARS-CoV-2 S-2P. WRAIR-5001-RBD is overlaid onto the RBD (dark gray surface) from one protomer. Side and top views are shown. **k** Sequence alignment of WRAIR-5001 with its precursor germline gene. CDRs are shaded grey, with residue numbering and CDR loops designated using the Kabat system. Residues interacting with the RBD are colored purple. Symbols *,:, and . denote identical, similar, and less similar residues, respectively. Source data are provided as a Source Data file.

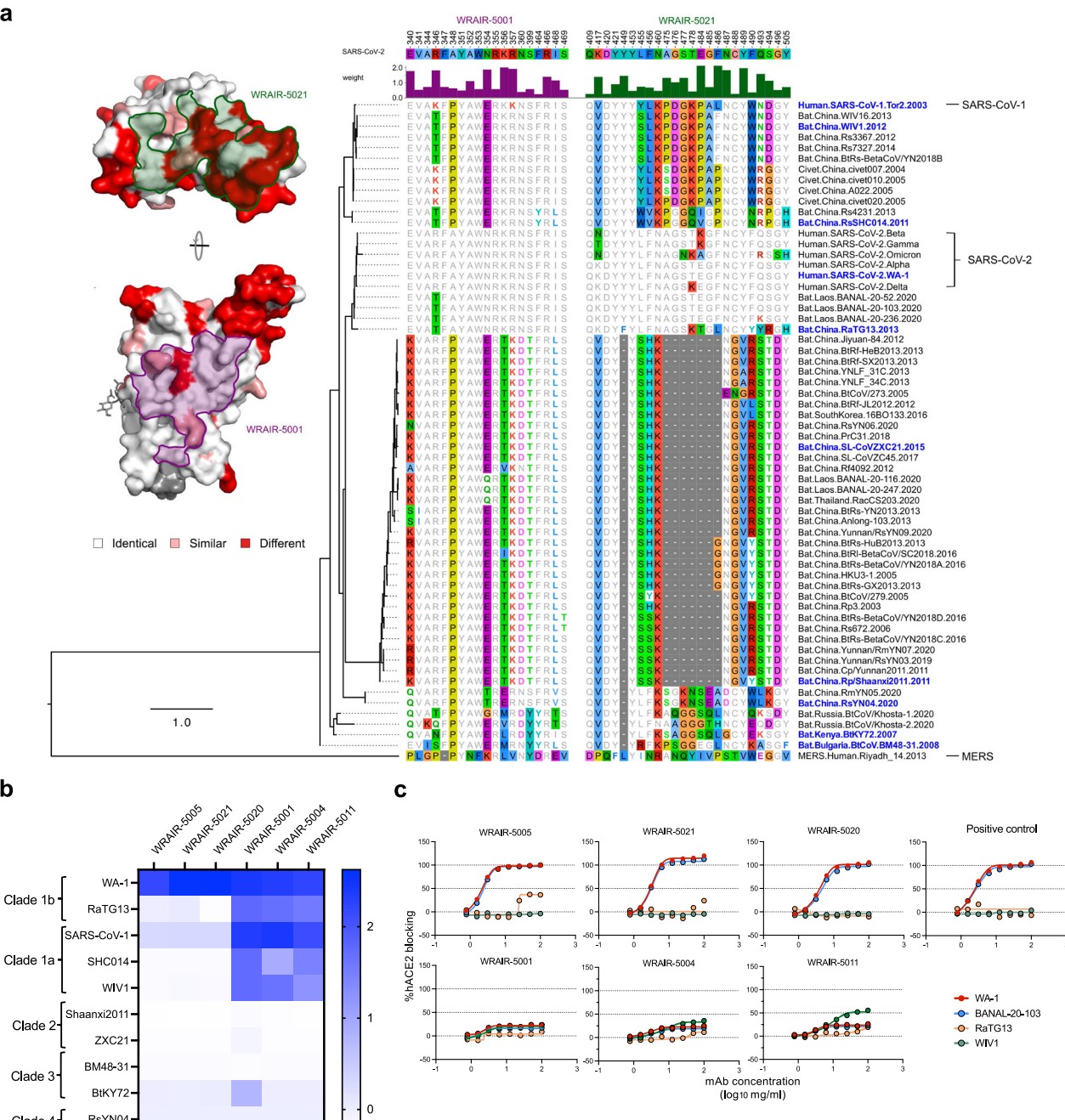

**Fig. 6 | Epitope conservation and mAb cross-reactivity. a** Structural and sequence analysis of the WRAIR-5021 and WRAIR-5001 footprints across sarbecoviruses. (Right) The epitope residues are numbered according to the Wuhan reference sequence; the strength of the interaction between the mAb and the RBD is indicated by the height and color of the histogram bars above the sequence alignment. Sequences are ordered based on their phylogenetic relationships based on a maximum likelihood phylogenetic tree derived from the RBD protein sequences. (Left*)* The RBD structure is shown in surface representation and depicts mutations between SARS-CoV-1 and SARS-CoV-2 in red; the WRAIR-5021 and WRAIR-5001 epitopes are outlined and labeled. (Right) Sequence alignment of mAb epitopes across sarbecoviruses. **b** The binding of mAbs was measured with a set of sarbecovirus RBDs using BLI to assess cross-reactivity. Heat-map represents the relative binding strengths for WRAIR-mAbs. Coloring legend indicating the relative binding strength is shown on the right. **c** ACE2 blocking activity of WRAIR RBD mAbs in a BLI-based assay. mAbs were assessed for their ability to block human ACE2 binding to selected sarbecovirus clade 1b or 1a RBDs. Source data are provided as a Source Data file.

WRAIR-5001 epitope exists between the class III and class V epitopes (Fig. 5f, g). Further analysis revealed that the light chain of the WRAIR-5001 overlaps with S309 epitope, a class-III mAb, and the heavy chain epitope extends along the WRAIR-2057 epitope (Fig. 5g), confirming the epitope binning assays (Supplementary Fig 1a and 6a, b). Structural mapping of the variant mutations on the RBD demonstrated that none of the mutant residues in the

Alpha, Beta, Gamma, Delta and Omicron BA.1 and BA.2 variants are within the epitope site of WRAIR-5001 mAb and therefore were not expected to impact the WRAIR-5001 binding and neutralization (Fig. 5h). This finding was verified by affinity assessment of WRAIR-5001 for select RBDs, all of which bound with low nanomolar or high picomolar affinity (Fig. 3e, Table S2). Our analysis revealed that even in the context of highly mutated variants, i.e.

Delta or Omicron, only a single mutation overlaps with the WRAIR-5001 epitope (G339D).

To further analyze the WRAIR-5001 epitope in the context of the SARS-CoV-2 prefusion stabilized Spike trimer, we modeled WRAIR-5001 onto the closed (all RBD down; PDB code: 6VXX), and 1- or 2-RBD up conformations (PDB codes: 7DWZ and 6X2B) (Fig. 5i, j)[51–53]. Structure superimposition demonstrated that the epitope for WRAIR-5001 is occluded by the adjacent S protomers when the RBD is in the "down" conformation while more accessible when RBD is in the open conformation, indicating the cryptic nature of this epitope in line with other mAbs in class III or class V. This type of antibody exhibits low SHM originating from the precursor germline genes in rhesus macaques, VH1-200*01 and VL3-36*01 (Fig. 5k).

**Epitope conservation analysis.** Structural and sequence analysis revealed that the WRAIR-5001 epitope is highly conserved between SARS-CoV-2 and SARS-CoV-1 RBDs with >85% sequence identity (Fig. 6a). Given this high sequence identity, we analyzed the conservation of the WRAIR-5001 epitope among SARS-CoV-2 VoC and other representative sarbecoviruses (Fig. 6a). Our sequence alignment and phylogenetic analysis demonstrated that the epitope for WRAIR-5001, is considerably conserved amongst sarbecoviruses; in contrast, we observed significant variations for the WRAIR-5021 epitope (Fig. 6a), consistent with other mAbs that target the hACE2 binding site.

Intrigued by the sequence conservation of the WRAIR-5001 epitope, we tested the antigenic cross-reactivity of all the macaque RBD-directed mAbs with a panel of sarbecovirus RBD molecules (Fig. 6b). Overall, the RBD-B mAbs (WRAIR-5001, WRAIR-5004, and WRAIR-5011) exhibited greater breadth than RBD-A mAbs (WRAIR-5005, WRAIR-5020, and WRAIR-5021), cross-reacting with diverse RBDs from sarbecovirus clades 1b and 1a, which utilize hACE2 for cell entry. We explored the ability of our RBD-A and RBD-B mAbs to block hACE2 binding in the presence of RBDs from diverse sarbecoviruses (Fig. 6c). Group A RBD mAbs only blocked hACE2 binding to BANAL-20-103 RBD, another clade 1b sarbecovirus, with equivalent activity to WA-1. Interaction of Group A WRAIR-5005 and -5021 with RaTG13 was minimal, limiting hACE2 inhibition. Group B RBD mAbs displayed modest inhibition potential to WA-1, but this was consistently maintained in the hACE2-blocking assays using BANAL-20-103, RaTG13, and WIV1 RBD molecules. Combined, these data indicate that SpFN-elicited antibodies cross-recognize clade 1a and clade 1b sarbecoviruses.

## Discussion

Considering the continued changing landscape of SARS-CoV-2 viral evolution, next-generation vaccines are needed for protection against COVID-19. Novel nanoparticle-based SARS-CoV-2 and pan-CoV vaccines in development show promising elicitation of potent and broad sarbecovirus neutralization responses[55–57], including protection against highly divergent VoC including Beta, Delta and Omicron[24]. Here, we detail the identification and characterization of antibodies elicited following immunization of rhesus macaques with SpFN/ALFQ, to understand the molecular mechanism of the broad neutralization observed in multiple pre-clinical animal models. We demonstrate that the polyclonal humoral response in the serum of SpFN-vaccinated macaques targets overlapping, but distinct, epitopes of the Spike when compared to a first-generation COVID-19 mRNA-LNP vaccine in macaques. We isolated mAbs targeting multiple regions of the immunogen including the ferritin nanoparticle, and Spike NTD, RBD and S2, and characterized the activity and cross-reactivity of these mAbs. The S-targeting mAbs included both neutralizing and non-neutralizing molecules which displayed high, yet differential, activity in multiple antibody effector assays dependent on the antigenic target, suggesting a role for the Fv in facilitating both neutralization and Fc effector functions as all isolated mAbs were cloned into the same rh-IgG1 backbone. Antibodies that facilitate Fc effector functions such as ADCP, from NTD and RBD mAbs, are found following SARS-CoV-2 infection, and may contribute towards protection[34,58].

Previous studies have shown that one of the principal components of vaccine-elicited SARS-CoV-2 humoral immunity are neutralizing RBD-targeting antibodies[59,60]. In the case of SpFN-elicited RBD-targeting mAbs, we divided them into two broad epitope categories targeting either (i) the hACE2-binding site (Group A RBD mAbs, associated with class I), or (ii) a conserved cryptic site (Group B RBD mAbs, associated with class III/V). We structurally characterized the SARS-CoV-2 WA-1 RBD in complex with WRAIR-5021 and WRAIR-5001, which represent RBD competition Groups A and B, respectively. The WRAIR-5021 epitope overlaps considerably and competes with the hACE2 binding site, and exhibits robust binding to RBDs from VoCs, including Delta, Omicron BA.1, and Omicron BA.4/5, with low nanomolar affinity was observed. However, potent neutralization was absent for most of the Omicron variants. Matching with the neutralization data, WRAIR-5021 showed robust protection in the K18 mouse model challenged using Delta. The WRAIR-5001 mAb, while having more modest neutralization potency, targeted a highly conserved epitope located between the class III and class V epitopes. This epitope is cryptic and only seen in the RBD-up conformation of Spike and has high structural and sequence conservation. WRAIR-5001 showed robust binding to a panel of clade 1a and 1b sarbecovirus RBD molecules with sub-nanomolar affinity and neutralized Delta, Omicron BA.1, Omicron BA.2, and SARS-CoV-1, but did not yield protection in the stringent K18 mouse model against a Delta VoC challenge, likely owing to the modest neutralization activity toward the Delta VoC.

This study has provided molecular detail the RBD-specific antibodies that are elicited following SpFN vaccination and explains the previously observed broad neutralizing and protective immune responses. Similar analysis using mAb-blocking assays with DH1041 and DH1047 showed how other nanoparticle-based vaccines elicited immune responses that are focused on epitopes consistent with WRAIR-5001 and WRAIR-5021[61]. This current study links vaccine-elicited rhesus macaque antibodies that recapitulate the properties and epitope-targeting of broadly neutralizing and potent mAbs found in humans.

As previously noted for nanoparticle-based vaccines, the difference in immune responses when compared to single-valence vaccines is likely due to multiple reasons including the repetitive nature of a displayed antigen, improved T-cell help, and increased antigen duration in lymph nodes. In numerous cases, this has been shown to generate improved quality of immune responses against multiple viral pathogens[29,31,55,62–68]. Adjuvant selection likely also enhances the overall adaptative immune responses observed after SpFN vaccination, as ALFQ has been shown to also elicit robust T cell stimulation[23,25,26]. Thus, vaccination with SpFN and ALFQ in combination leads to strong engagement of both the humoral and cellular components of the immune system, including the ability of ALFQ to stimulate CD4 T cell help, leading to the development of mature, vaccine-responsive B cells. The further development of these SARS-CoV-2 nanoparticle vaccines and characterization in humans is important, with the Phase I clinical evaluation of SpFN currently underway, with clinical and immunological data in preparation [NCT04784767]. Preliminary data from this trial indicates the SpFN/ALFQ vaccination in humans recapitulates findings from the pre-clinical animal model studies and stimulates robust CD4 T cell-mediated immunity concurrently with peak neutralizing titers. Understanding the context of SpFN-vaccine elicited responses in context of prior coronavirus exposure in humans will yield insight into the effectiveness of these responses to protect against current VoC. In addition, this data will further speak to the ability of these vaccines to generate immune coverage against SARS-CoV-2 emerging VoC and potential spillover events from distinct sarbecoviruses.

The mAbs identified in this study, while a relatively small number and from one representative animal, appear to provide a reasonable representation of the antibodies present in SpFN-immunized macaque serum, and defined the RBD-specific antibody molecular recognition of neutralization and cross-recognition of sarbecoviruses elicited by SpFN vaccination. Although we did not further define the molecular recognition of NTD- and S2-specific mAbs beyond epitope mapping, these sites were also important targets of SpFN-elicited antibodies. Efforts to isolate and identify the targets of neutralization elicited in the ongoing Phase I clinical trial of SpFN/ALFQ will confirm and expand upon these findings. We are encouraged that neutralizing epitopes targeted by the mAbs isolated in this study overlap with antibody targets elicited in SARS-CoV-2 convalescence, suggesting that SpFN will confer protection beyond animal models. Understanding antibody epitope specificities and functional characteristics of cross-reactive antibodies following vaccination will further aid the development of antiviral countermeasures for SARS-CoV-2 and potential future CoV pandemic pathogens.

## Methods

### Sorting of SARS-CoV-2-positive B cells
Cryopreserved PBMCs from naïve or SpFN vaccinated rhesus macaques were thawed in warm media containing benzonase, then washed with PBS and stained for viability using the Aqua Live/Dead stain (ThermoFisher) and Fc receptors blocked using normal mouse IgG (Invitrogen #02-6502). After centrifuging, cells were then stained for 30 min at RT for SARS-CoV-2 and SARS-CoV-1 reactive B cells by incubating with SpFN and SpFN[1] particles, followed by a wash with staining buffer (1x PBS containing with FCS and NaN$_3$). Cells were then incubated at RT for 30 min with fluorochrome-conjugated secondary antibodies specific to the S protein of either SARS-CoV-2 (WRAIR-2054-APC) or SARS-CoV-1 (mm02-PE, SinoBiological #40150-MM02). Finally, after another wash, cells were incubated at RT for 30 min with a cocktail of phenotyping antibodies including CD3 BV510 (BD Biosciences Cat# 740187, Lot# 0261796), CD4 BV510 (BD Biosciences Cat# 562970, Lot# 0177290), CD14 BV510 (BioLegend Cat# 301842, Lot# B264335), and CD16 BV510 (Biosciences Cat# 563830, Lot# 1307468) as dump channel markers, and CD19 PE-Cy5 (Beckman Coulter Cat# IM2643U Lot# 200100), IgG PE-Cy7 (BioLegend Cat# 410721 Lot# M1310G05), IgM BV650 (Biolegned Cat# 314525 Lot# B294373). SARS-CoV-2 RBD, and NTD (ThermoFisher) that had been biotinylated, tetramerized and conjugated to streptavidin-PE were also included in this step. Viable, dump channel-, IgG+, IgM-, CD19$^+$ B cells that were antigen positive to either a SpFN particle or tetramerized subdomain, or a combination, were single-cell sorted into PCR plates containing lysis buffer composed of murine RNAse inhibitor (New England Biolabs), dithiothreitol (DTT), SuperScript III First Strand Buffer (Thermo-Fisher), Igepal (Sigma), and carrier RNA (Qiagen) at one cell per well using a FACS ARIA (Becton Dickinson) and stored at -80 °C until subsequent reverse transcription. Analysis was performed using FlowJo 10 (BD Bioscience).

### Antibody sequencing and production
RNA from single B cells was reverse-transcribed using random hexamers and the SuperScriptIII kit (ThermoFisher). Antibody V (D) J genes were amplified from the cDNA by nested PCR, using the HotStar Taq DNA Polymerase kit (Qiagen) using a combination of primer sets and methods described previously[69]. V(D)J gene assignment, somatic hypermutation and CDR3 determinations were performed using IgBlast. Antibody variable regions were synthesized and cloned (Genscript) into CMVR expression vectors (NIH AIDS reagent program) between a murine Ig leader (GenBank DQ407610) and the constant regions of human IgG1 (GenBank AAA02914), Igκ (GenBank AKL91145) or Igλ (GenBank AAA02915). Antibodies were first expressed using a 24-well plate format by co-transfecting plasmids encoding paired heavy and light chains into Expi293F cells (ThermoFisher) according to the manufacturer's instructions. After 5 days, clarified cell culture supernatants were screened for neutralization in the pSV assay and binding to SARS-CoV-2 antigens using a bead-based multiplex assay (see above). Positive hits from the supernatant screen for neutralization activity and/or binding were scaled up by transfecting 30 ml cultures of Expi293F cells as indicated above. Monoclonal antibodies were purified 4 to 5 days post-transfection using AmMag Protein A magnetic beads and the AmMagSA purification system (Genscript), according to the manufacturer's recommendations. Purified mAbs were buffer exchanged into Phosphate-Buffered Saline (PBS). The purity and stability of monoclonal antibodies were assessed by SDS-PAGE and Coomassie staining in both reducing and non-reducing conditions. Control antibodies were all expressed as human IgG1 and purified from Expi293F cells, as described above.

### Fab production
Freshly purified WRAIR IgGs in PBS buffer (pH 7.4) were mixed with Lys C protease (New England Biolabs) at 1:2000 (w:w) ratio. Reaction was allowed to proceed for 2–3 h in a water bath incubator at 37 °C. Digestion was assessed by SDS-PAGE and upon completion, the reaction mixture was passed through Protein-A beads (Cytiva) three times and the final flow through was assessed by SDS-PAGE for purity.

### Production of recombinant proteins
Recombinant SARS-CoV-2 proteins RBD (318-514), NTD (1-290) and S1 (1-665) were made from a synthesized full-length Spike sequence (Genscript) from strain USA/IL1/2020 (GenBank # MN988713) and were cloned with C-terminal AviTag and poly-histidine tags into the CMVR vector under the bovine prolactin leader sequence. The coding sequence for the SARS-CoV-2 (Genbank # MN908947) stabilized trimer (S-2P) was a generous gift from Jason McLellan[70]. The S-2P sequence was subcloned into the pCMVR vector with C-terminal Avi-Tag and poly-histidine tags. Four additional stabilizing mutations were added using the Quickchange multisite-directed mutagenesis kit (Agilent) to make the HexaPro variant with improved stability[71], referred to as stabilized S trimer throughout the manuscript. SARS-CoV-2 RBD constructs (331 - 527) were also modified to incorporate a N-terminal hexa-histidine tag were derived from the Wuhan-Hu-1 strain genome sequence (GenBank # MN9089473). Subsequent RBD VoC with point mutations were generated using a modified QuikChange site-directed mutagenesis protocol (Agilent). A S-2P construct derived from SARS-CoV-1 was generated as previously described[72]. Spike probes were expressed and biotinylated as previously described[73], with mutations for B.1.1.7, B.1.351, P.1 and other variants added by Quik-Change site-directed mutagenesis. Mutated residues were as follows: B.1.1.7 (69-70del, Y144del, N501Y, A570D, D614G, P681H, T718I, S982A, D1118H), B.1.351 (L18F, D80A, D215G, 241-243del, R246I, K417N, E484K, N501Y, D614G, A701V, E1195Q), P.1 (L18F, T20N, P26S, D138Y, R190S, K417T, E484K, N501Y, D614G, H655Y,T1027I), B.1.427/429 (W152C, L452R, D614G), B.1.526a (T95I, D253G, S477N, D614G, A701V) and B.1.526b (T95I, D253G, E484K, D614G, A701V). hACE2-Ig, a fusion protein made by connecting the human hACE2 (Q9BYF1) extracellular domain (residues 19-611) to the constant domain of a human IgG1 was expressed and purified as described above for antibodies. All proteins were produced transiently from Expi293F or FreeStyle 293F (stabilized trimer) cells (both ThermoFisher) and purified from cell culture supernatants using Ni-NTA (Qiagen) affinity. The stabilized trimer was further purified by gel filtration on an ENrich SEC 650 column (Bio-Rad)

### Multiplex antibody binding assay
A high-throughput bead-based antibody binding assay was performed as previously described[74,75] with modifications to adapt to sarbecovirus antigens. Briefly, heat-inactivated serum from rhesus macaques that

were vaccinated twice with 50 μg of SpFN or PBS, or purified mono-clonal antibodies, were diluted and loaded into 384-well assay plates by use of a Biomek NXP® automated liquid handler (Beckman Coulter). A cocktail of 11 sarbecovirus antigens and 1 control proteins (HIV-1 antigens), obtained commercially (SinoBiological) or internally produced (see below), spanning Spike S1 and S2 domains for SARS-CoV-2 or SARS-CoV-1 were covalently coupled to uniquely coded magnetic microspheres (Luminex) per manufacturer's protocol and added to the plate in a final volume of 50 μl/well. Following a 2 h incubation with vigorous shaking, microspheres were washed using a magnetic 384-well automated plate washer (Bio-Tek) to remove unbound sample. Microspheres were then resuspended with 0.5 μg ml$^{-1}$ mouse anti-human IgG-PE (Southern Biotech), vortexed for 1 min with a microplate vortex at 3,000 rpm, sonicated for 1 min and then incubated with vigorous shaking for 1 h. A final wash removed unbound detection reagent, and microspheres were resuspended in 40 μl sheath fluid (Luminex). Data was collected on a Bio-Plex®3D Suspension Array system (Bio-Rad) running xPONENT® v.4.2 (Luminex). Signal to Noise (S/N) ratio were calculated by the dividing the MFI for each sample by either Ig-depleted healthy serum or a negative control antibody (MZ4) according to the type of sample analyzed.

### SARS-CoV-2 pseudovirus neutralization assay

SARS-CoV-2 pseudovirions (pSV) were produced by co-transfection of HEK293T/17 cells with a pcDNA3.1 encoding SARS-CoV-2 S and an HIV-1 NL4-3 luciferase reporter plasmid (pNL4-3.Luc.R-E-, NIH AIDS Reagent Program). The S expression plasmid sequence was derived from the Wuhan Hu-1 strain or other indicated strain (Genscript) (GenBank # NC_045512), which is also identical to the IL1/2020 and WA1/2020 strains. The S expression plasmid sequence was also codon optimized and modified to remove the last 18 amino acids of the cytoplasmic tail to improve S incorporation into the pseudovirions and thereby enhance infectivity. S expression plasmids for current SARS-CoV-2 VoC were similarly codon optimized, modified and included the following mutations compared to WA-1: Beta/B.1.351 (D80A, D215G, del 241-243, K417N, E484K, N501Y, D614G, A701V,), Delta/B.1.617.2 (T19R, E156G, del 157-158, L452R, T478K, D614G, P681R, D950N), Omicron BA.1 (A67V, Δ69/70, T95I, G142D, del 143-145, N211I, del 212R214, insG339D S371L, S373P, S375F, K417N, N440K, G446S, S477N, T478K, E484A, Q493R, G496S, Q498R, N501Y, Y505H, T547K, D614G, H655Y, N679K, P681H, N764K, D796Y, N856K, Q954H, N969K, L981F), Omi-cron BA.2 (T19I, L24-, P25-, P26-, A27S, G142D, V213G, G339D, S371F, S373P, S375F, T376A, D405N, R408S, K417N, N440K, S477N, T478K, E484A, Q493R, Q498R, N501Y, Y505H, D614G, H655Y, N679K, P681H, N764K, D796Y, Q954H, N969K), and Omicron BA.5 (T19I, del L24, del P25, del P26, A27S, del H69, del V70, G142D, V213G, G339D, S371F, S373P, S375F, T376A, D405N, R408S, K417N, N440K, L452R, S477N, T478K, E484A, F486V, Q498R, N501Y, Y505H, D614G, H655Y, N679K, P681H, N764K, D796Y, Q954H, N969K) from Genscript. A D614G var-iant was also made from the Wuhan Hu-1 construct using the Q5 site-directed mutagenesis kit (NEB). In addition, a codon-optimized S expression plasmid encoding SARS-CoV-1 (Sino 1-11, GenBank # AY485277) was generated that incorporated a 28 amino acid C-terminal deletion to improve infectivity[76]. Virions pseudotyped with the vesicular stomatitis virus (VSV) G protein were used as control. Infectivity and neutralization titers were determined using hACE2-expressing HEK293 target cells (Integral Molecular) in a semi-automated assay format using robotic liquid handling (Biomek NXp Beckman Coulter). Samples were diluted 1:40 in growth medium and serially diluted, then 25 μl/well was added to a white 96-well plate. Purified mAbs started at a concentration of 1 mg ml$^{-1}$. An equal volume of diluted SARS-CoV-2 pSV was added to each well and plates were incubated for 1 h at 37 °C. Target cells were added to each well (40,000 cells/well) and plates were incubated for an additional 48 h. RLUs were measured with the EnVision Multimode Plate Reader (Perkin Elmer)

using the Bright-Glo Luciferase Assay System (Promega). Neutraliza-tion dose–response curves were fitted by nonlinear regression using the LabKey server, and the final titers are reported as the reciprocal of the dilution of serum necessary to achieve 50% neutralization (ID$_{50}$, 50% inhibitory dose or IC$_{50}$, 50% inhibitory concentration) and 80% neutralization (ID$_{80}$, 80% inhibitory dose or IC$_{80}$, 80% inhibitory con-centration). Assay equivalency was verified by participation in the SARS-CoV-2 Neutralizing Assay Concordance Survey (SNACS) run by the Virology Quality Assurance Program and External Quality Assur-ance Program Oversite Laboratory (EQAPOL) at the Duke Human Vaccine Institute, sponsored through programs supported by the National Institute of Allergy and Infectious Diseases, Division of AIDS.

**Authentic SARS-CoV-2 variant and SARS-CoV-1 neutralization assay.** Assays were performed as previously described[23] (34914540), with SARS-CoV-2 viruses USA-WA1/2020 (WA-1), USA/CA_CDC_5574/2020 (B1.1.7), hCoV-19/South Africa/KRISP-EC-K005321/2020 (B.1.351), hCoV-19/Japan/TY7-503/2021, and hCoV-19/USA/PHC658/2021 (B.1.617.2) obtained from BEI Resources (National Institute of Allergy and Infectious Diseases, NIH) and propagated for one passage using Vero clone E6 cells. Virus infectious titer was determined by an end-point dilution and cytopathic effect (CPE) assay on Vero-E6 cells. An endpoint dilution microplate neutralization assay was performed to measure the neutralization activity of macaque serum samples. In brief, serum samples were heat-inactivated and subjected to succes-sive threefold dilutions starting from 1:50. Triplicates of each dilution were incubated with SARS-CoV-2 at a multiplicity of infection of 0.1 in Eagle's minimum essential medium with 7.5% inactivated fetal calf serum for 1 hour at 37 °C. After incubation, the virus-antibody mixture was transferred onto a monolayer of Vero-E6 cells grown overnight. The cells were incubated with the mixture for about 70 hours. CPE of viral infection was visually scored for each well in a blinded fashion by two independent observers. The results were then reported as the percentage of neutralization at a given sample dilution. A SARS-CoV-1 authentic plaque reduction virus neutralization assay was performed similarly to previously described[23] (58), with the following modifica-tions. The starting dilution of serum was 1:5, and about 100 plaque-forming units of virus were used for virus and serum incubation. The overlay used after virus adsorption was Dulbecco's modified Eagle's medium (Gibco) containing 2% FBS and 20% methylcellulose. Plates were then incubated for 5 days, and after crystal violet staining, the washing step used water. Plaques were graded as follows: about 25 plaques/25% monolayer damage (MD; ±); about 50 plaques/50% MD (+); about 75 plaques/75% MD (++); and about 100 plaques/100% MD (+++). All negative control wells were solid monolayers.

**Measurements of antibody Fc effector functions using recombinant proteins.** ADCP was measured as previously described[77]. Briefly, bio-tinylated SARS-CoV-2 S stabilized trimer was incubated with red streptavidin-fluorescent beads (Molecular Probes) for 2 h at 37 °C. Ten μl of a 100-fold dilution of beads–protein mixture was incubated for 2 h at 37 °C with 100 μl of monoclonal antibodies diluted at 5 μg ml$^{-1}$ before addition of THP-1 cells (20,000 cells per well; Millipore). After 19 h incubation at 37 °C, the cells were fixed with 2% formaldehyde solution and fluorescence was evaluated on a LSRII flow cytometer (BD Bioscience). The phagocytic score was calculated by multiplying the percentage of bead-positive cells by the geometric mean fluorescence intensity (MFI) of the bead-positive cells and dividing by 10$^4$.

### Measurements of antibody Fc effector functions using cell surface-expressed Spikes

**Opsonization.** SARS-CoV-2 S-expressing FreeStyle 293 F cells were generated by transfection with linearized plasmid encoding a codon-optimized full-length SARS-CoV-2 S protein matching the amino acid sequence of the IL1/2020 isolate (GenBank # MN988713). Stable

transfectants were single-cell sorted and selected to obtain a high-level Spike surface expressing clone (293F-Spike-S2A). 293F-Spike-S2A cells were incubated with 100 µl of monoclonal antibodies diluted at 5 µg ml⁻¹ for 30 min at 37 °C. Cells were washed twice and stained with anti-human IgG PE (Southern Biotech). Cells were then fixed with 4% formaldehyde solution and fluorescence was evaluated on a LSRII (BD Bioscience).

Trogocytosis was measured using a previously described assay[78]. Briefly, SARS-CoV-2 Spike–expressing Expi293F cells were stained with PKH26 (Sigma-Aldrich). Cells were then washed with and resuspended in R10 media. Cells were then incubated with monoclonal antibodies diluted at 5 µg ml⁻¹ for 30 min at 37 °C. Effector peripheral blood mononuclear cells were next added to the R10 media at an effector to target (E:T) cell ratio of 50:1 and then incubated for 5 h at 37 °C. After the incubation, cells were washed, stained with live/dead aqua fixable cell stain (Life Technologies) and CD14 APC-Cy7 (clone MɸP9) for 15 min at RT, washed again, and fixed with 4% formaldehyde (Tousimis) for 15 min at RT. Fluorescence was evaluated on an LSRII flow cytometer (BD Biosciences). Trogocytosis was evaluated by measuring the PKH26 mean fluorescence intensity of the live CD14⁺ cells.

**CD16 reporter assay (ADCC).** WT Spike-CEM cells were plated at 100,000 per well in round bottom 96-well plates and incubated with 5 µg ml⁻¹ of mAbs for 30 min at 4 °C. Cells were washed and 200,000 Jurkat-Lucia NFAT-CD16 cells (Invivogen) were added to each well in 100 µl of IMDM 10% FBS. The cells were then centrifuged for 1 min at low speed and co-cultured for 24 h at 37 °C. Fifty µl of Quanti-Luc was added to 20 µl of co-culture supernatant and luminescence was measured immediately on a luminometer (2104 Multilabel reader, PerkinElmer).

**Epitope binning**
Epitopes of the NTD and RBD mAbs were first mapped by binding competition against a set of characterized control antibodies (RBD) using Biolayer interferometry (BLI) on an Octet RED96 instrument (FortéBio) similar to what was reported previously[34]. Avi-tagged recombinant NTD and RBD proteins, biotinylated with the BirA biotinylation kit (Avidity), were diluted to 2.5 and 1 µg ml⁻¹, respectively, in kinetic buffer (0.1% [w/v] bovine serum albumin [BSA], 0.02% [v/v] Tween-20 in PBS; FortéBio) and loaded onto Streptavidin (SA) sensors (FortéBio) for 250 s, to reach ~50% of the sensor maximum binding capacity. Loaded biosensors were immersed into wells containing the first competing antibody at 100 nM for 900 s to saturate all binding sites. Next, biosensors were dipped into wells containing the second antibody, in the presence of the first competing antibody (all at 100 nM), and binding was measured after 900 s of association. Residual binding signal of the second antibody was expressed as a percentage of the maximum binding signal obtained in absence of the first competing antibody, ran in parallel. Residual binding signal was further corrected for any increase in signal obtained with the first competing antibody alone. Antibodies were defined as competing when binding signal of the second antibody was reduced to less than 25% of its maximum binding capacity and non-competing when binding was greater than 50%. Intermediate competition was defined by binding levels of 25-50%. Control antibodies representing RBD-A, RBD-B and RBD-C, as described previously[34], were WRAIR-2125, WRAIR-2063 and WRAIR-2151, respectively. Control antibodies representing NTD-A, NTD-B and NTD-C, as described previously[34], were WRAIR-2025, WRAIR-2137 and WRAIR-2054, respectively. The same approach was used to assess binding competition between NTD and RBD antibodies within the stabilized S trimer. hACE2-Ig was used like an antibody to assess the ability of NTD and RBD antibodies to block hACE2 binding to the S trimer.

**Affinity binding assays**
Real-time interactions between purified SARS-CoV-2 proteins and antibodies were monitored on an Octet RED96 instrument (FortéBio). For affinity measurement, mAbs were immobilized onto Anti-human IgG Fc capture (AHC) biosensors (Sartorius). The baseline was established in PBS. Loaded biosensors were dipped into wells containing serial dilutions of RBD (starting from 500 nM) for 180 s. Complexes were then allowed to dissociate in PBS for 300 s. After reference subtraction, apparent binding kinetic constants were determined, from at least 4 concentrations of RBD, by fitting the curves to a 1:1 binding model using the Data analysis software 12.0 (FortéBio). To assess binding to a panel of RBD mutants, HIS1K biosensors (FortéBio) were equilibrated in assay buffer (PBS) for 15 s before loading of His-tagged SARS-CoV-2 RBD, VoC RBDs, or SARS-CoV-1 RBD (30 µg ml⁻¹ diluted in PBS) for 100 s. Immobilized RBD proteins were then dipped in antibodies (30 µg ml⁻¹ diluted in PBS) for 180 s followed by dissociation for 60 s. To assess binding to the panel of S mutants, biotinylated probes were loaded on SA biosensors (FortéBio) and subsequently dipped into antibodies (30 µg ml⁻¹ diluted in 1X kinetic buffer) for 450 s followed by a 120 s dissociation step. Binding responses were measured at the end of the association step using the Data analysis software 10.0 (FortéBio). hACE2-RBD competition assays were carried out as follows: SARS-CoV-2 RBD (30 µg ml⁻¹ diluted in PBS) was immobilized on HIS1K biosensors (FortéBio) for 220 s. Test antibodies were allowed to bind for 200 s, followed by baseline equilibration (30 s), and then incubation with hACE2 protein (30 µg ml⁻¹) for 120 s. Percent inhibition (PI) of RBD binding to hACE2 by antibodies was determined using the equation: PI = [(hACE2 binding following RBD-antibody incubation)) / (hACE2 binding)] × 100. Antibody concentration was titrated from 100 µg ml⁻¹ by serial two-fold dilutions. All assays were performed at 30 °C with agitation set at 1000 rpm.

**Epitope mapping of antibodies by alanine scanning**
Epitope mapping was performed essentially as described previously[79] using SARS-CoV-2 (strain Wuhan-Hu-1) S protein RBD shotgun mutagenesis mutation libraries, made using a full-length expression construct for S protein. 184 residues of the RBD (between S residues 335 and 526) were mutated individually to alanine, and alanine residues to serine. Mutations were confirmed by DNA sequencing, and clones arrayed in 384-well plates, one mutant per well. The binding of mAbs to each mutant clone in the alanine scanning library was determined, in duplicate, by high-throughput flow cytometry. Each S protein mutant was transfected into HEK-293T cells and allowed to express for 22 h. Cells were fixed in 4% (v/v) paraformaldehyde (Electron Microscopy Sciences), and permeabilized with 0.1% (w/v) saponin (Sigma-Aldrich) in PBS plus calcium and magnesium (PBS++) before incubation with mAbs diluted in PBS++, 10% normal goat serum (Sigma), and 0.1% saponin. MAb screening concentrations were determined using an independent immunofluorescence titration curve against cells expressing wild-type S protein to ensure that signals were within the linear range of detection. Antibodies were detected using 3.75 µg ml⁻¹ of AlexaFluor488-conjugated secondary antibody (Jackson ImmunoResearch Laboratories) in 10% normal goat serum with 0.1% saponin. Cells were washed three times with PBS++/0.1% saponin followed by two washes in PBS and mean cellular fluorescence was detected using a high-throughput Intellicyte iQue flow cytometer (Sartorius). Antibody reactivity against each mutant S protein clone was calculated relative to wild-type S protein reactivity by subtracting the signal from mock-transfected controls and normalizing to the signal from wild-type S-transfected controls. Mutations within clones were identified as critical to the mAb epitope if they did not support reactivity of the test mAb, but supported reactivity of other SARS-CoV-2 antibodies. This counter-screen strategy facilitates the exclusion of S mutants that are locally misfolded or have an expression defect.

## Serum antibody epitope mapping

Serum antibody epitope mapping competition assays were performed as previously described[36], using the Biacore 8K+ surface plasmon resonance system (Cytiva). Briefly, anti-histidine antibody was immobilized on Series S Sensor Chip CM5 (Cytiva) via primary amine coupling using a His capture kit (Cytiva). His-tagged SARS-CoV-2 S protein containing 2 proline stabilization mutations (S-2P) was then captured on the active sensor surface. Human IgG monoclonal antibodies (mAbs) used for these analyses include: S2-specific mAbs S652-112, and S2P6; NTD-specific mAbs 4-8, S652-118, 5-7, and N3C; S1-specific mAb A20-36.1; and RBD-specific mAbs B1-182, CB6, A20-29.1, A19-46.1, LY-COV555, A19-61.1, S309, A23-97.1, A19-30.1, WRAIR-5001, WRAIR-5011, A23-80.1, and CR3022. Negative control antibody or competitor mAb was injected over both active and reference surfaces, followed by subsequent injection of non-human primate sera (diluted 1:100). Active and reference sensor surfaces were regenerated following each analysis cycle. Prior to analysis, sensorgrams were aligned to Y (Response Units) = 0, beginning at the serum association phase, using Biacore 8K Insights Evaluation Software (Cytiva). Relative "analyte binding late" report points (RU) were collected and used to calculate percent competition (% C) using the following formula: $\%\,C = [1 - (100 * ((RU$ in presence of competitor mAb) / (RU in presence of negative control mAb))]. Absolute competition ($\Delta$RUs) was calculated with the following formula: $\Delta$RUs = [(RUs in presence of negative control mAb) – (RUs in presence of competitor mAb)]. Results are reported as percent competition or absolute competed RUs and statistical analysis was performed using unpaired, two-tailed t-test (GraphPad Prism v.8.3.1). Assays were performed in duplicate, with average data point represented on corresponding graphs.

## In vivo protection studies in K18-hACE2 transgenic mice

All research in this study involving animals was conducted in compliance with the Animal Welfare Act, and other federal statutes and regulations relating to animals and experiments involving animals and adhered to the principles stated in the Guide for the Care and Use of Laboratory Animals, NRC Publication, 1996 edition. The research protocol was approved by the Institutional Animal Care and Use Committee of the Trudeau Institute. K18-hACE2 transgenic mice were obtained from Jackson Laboratories (Bar Harbor, ME). Mice were housed in the animal facility of the Trudeau Institute and cared for in accordance with local, state, federal, and institutional policies in a National Institutes of Health American Association for Accreditation of Laboratory Animal Care-accredited facility. For the prophylactic protection studies, on day −1, groups of 13 K18-hACE2 mice (8-10 weeks of age) were injected intravenously with the purified antibodies at the indicated dose of 10 mg/kg. On study day 0, all mice were inoculated with $1.25 \times 10^4$ PFU of SARS-CoV-2 Delta B.1.617.2 via intranasal instillation, a challenge dose determined from a previous study[80]. All mice were monitored from study day 0 to study day 1, with body weight measurements taken every day, twice daily, every 12 h, out to day 10. Mice were euthanized if they displayed any signs of pain or distress as indicated by the failure to move after stimulated or inappetence, or if mice had greater than 25% weight loss compared to their study day 0 body weight. From each group, a subset (5) of mice, were sacrificed 2 days after the challenge for the determination of infectious virus titers in the lower respiratory tract (from bronchoalveolar lavage and lung tissue) using a PRNT assay.

## Evaluation of escape and selection of virus variants

For the evaluation of antibody escape ability, and generation of putative antibody escape S variants, a previously described chimeric recombinant VSV derivative (rVSV/SARS-CoV-2/GFP2E1) that encodes a SARS-CoV2 S protein in place of VSV-G, recapitulating the neutralization properties of authentic SARS-CoV-2[81], was prepared and passaged to generate diversity. Then, rVSV/SARS-CoV-2/GFP2E1 populations containing $10^6$ infectious units were incubated with individual antibodies (at 1.25, 2.5, 5 or 10 μg ml⁻¹ final concentration) or 1:1 mixtures of two antibodies (5 μg ml⁻¹ of each antibody) for 1 h at 37 °C. Then, the virus-antibody mixtures were incubated with $5 \times 10^5$ 293 T/hACE2cl.22 cells in 6-well plates. Two days later, supernatants were harvested from these passage 1 cultures, and a 100 μl aliquot of the cleared supernatant was incubated with the same concentration of antibodies and then used to infect $5 \times 10^5$ 293 T/hACE2cl.22 cells in 6-well plates, as before. After a second passage, infectious rVSV/SARS-CoV-2/GFP2E1 titers were measured in the passage 2 supernatants to indicate escape or lack thereof from the neutralizing antibodies. Titers were measured by inoculating 293 T/hACE2cl.22 cells in 96-well plates with serially diluted supernatant, and determining the number of infected cells by FACS, 16 h later.

For passage 2 cultures in which clear escape was observed, as evidenced by the appearance of numerous GFP-positive cells, RNA was isolated from aliquots of supernatant containing selected viral populations using NucleoSpin 96 Virus Core Kit (Macherey-Nagel). The purified RNA was subjected to reverse transcription using random hexamer primers and SuperScript VILO cDNA Synthesis Kit (Thermo Fisher Scientific). The cDNA was amplified using KOD Xtreme Hot Start DNA 396 Polymerase (Millipore Sigma) flanking the S encoding sequences. The PCR products were gel-purified and subjected to bulk Sanger-sequencing.

## Identification of ferritin reactive antibodies by enzyme linked immunosorbent assay

96-well Immulon "U" Bottom plates were coated with 1 μg ml⁻¹ of ferritin proteins in Dulbecco's Phosphate buffered saline, pH 7.4 (PBS). Plates were incubated at 4 °C overnight. After 30 min of blocking with blocking buffer (Dulbecco's PBS containing 0.2% bovine serum albumin, pH 7.4), at RT, the plates were washed 3x with wash buffer (Dulbecco's PBS containing 0.05% Tween 20, pH 7.4). Antibodies were serially diluted 5-fold in sample buffer (Dulbecco's PBS containing 0.2% bovine serum albumin and 0.05% Tween 20, pH 7.4), or at a single concentration as indicated, and added to duplicate wells. The plates were incubated for 1 h at RT. The plates were then washed 4 times with wash buffer. Horseradish peroxidase (HRP)-conjugated goat anti-human IgG, gamma chain specific antibody (Sigma) was added and incubated at RT for 30 min, followed by 4 washes with wash buffer and one wash with PBS. For development, the substrate mixture from TMB Substrate set (Biolegend) was added and incubated for 10 min, before the addition of the Stop Solution for TMB Substrate (Biolegend). Absorbance (A) was measured at 450 nm or 650 nm as indicated, using an ELISA reader iD3 (Molecular Devices, San Jose, CA).

## X-ray crystallography and structure analysis

WRAIR-5001-RBD (10 mg ml⁻¹) and WRAIR-5021-RBD (9 mg ml⁻¹) complexes were screened for crystallization conditions using an Art Robbins Gryphon crystallization robot, 0.2 μl drops, and a set of 1200 conditions. Crystal drops were observed daily using a Jan Scientific UVEX-PS with automated UV and brightfield drop imaging. Initial crystallization conditions were optimized manually by mixing protein and reservoir solutions in 1:1 (v:v) ratios. Crystals used for data collection grew in the following crystallization conditions: WRAIR-5001-RBD complex: 0.2 M Sodium chloride, 0.1 M Phosphate-citrate pH 4.5, 20% PEG 8,000 and WRAIR-5021-RBD complex: 0.2 M Sodium malonate pH 7.0, 20% w/v Polyethylene glycol 3,350.

Diffraction data for the WRAIR-5001-RBD and WRAIR-5021-RBD complexes were collected at Advanced Photon Source (APS), Argonne National Laboratory beamline 24-ID-E and measured using a Dectris Eiger 16 M PIXEL detector to a final resolution of 4.3 Å and 2.5 Å, respectively. Diffraction data indexing, integration, and scaling were carried out using the XDS-GUI[82]. Data collection statistics are reported in Table S2.

All the crystal structures described in this study were solved by molecular replacement using PHASER, and iterative model building, and refinement were performed in COOT and Phenix[83–85]. Phenix xtriage was used to analyze all the scaled diffraction data output from HKL2000 and XDS. Primarily, data was analyzed for measurement value significance, completeness, asymmetric unit volume, and possible twinning and/or pseudotranslational pathologies. To determine the structure of the Fab-SARS-CoV-2 RBD structures, we used the previously reported crystal structure of SARS-CoV-2 RBD. The heavy chain variable domain ($V_H$) of A17 mAb (PDB code: 3ZL4) and light chain variable domain ($V_L$) of 20350 mAb (PDB code: 5CZV) were used as the search models for WRAIR-5021. The heavy chain of antibody CH235UCA (PDB code: 6UDA) and light chain of 059-152-Fv (PDB code: 5XWD) were combined and used as the search model for WRAIR-5001 mAb. Search models for Fabs, were divided into Fv and Fc domains to carry out the molecular replacement searches. This approach was critical in finding solutions for all complexes. All structures were refined using Phenix.refine with positional, global isotropic B-factor refinement and defined TLS groups. Manual model building was performed in COOT. The Ramachandran plot as determined by MOL-PROBITY showed > 96% of all residues in favored regions and ~4% of all residues in the allowed regions for the WRAIR-5001-RBD complex. For the WRAIR-5021-RBD complex >96% residues were located in the favored and allowed regions. Data collection and refinement statistics are reported in Table S2. Interactive surfaces were analyzed using PISA and are provided in Tables S4, S5. Structure figures were prepared using PyMOL (The PyMOL Molecular Graphics System, Version 2.1 Schrodinger, LLC). Software used in this work was curated by SBGrid[86].

The weight of each epitope sites used in the epitope conservation analysis was calculated based on antigen-antibody interactions in the determined antigen-RBD complexes as previously described[87,88]. Conservation was determined based on a previously selected set of representative sarbecovirus sequences as in Chen, et al.[88].

### Statistical analysis

Neutralization is the geometric mean of the $IC_{50}$ values calculated using 5-parameter logistic regression from at least two-independent experiments performed in triplicates (R package nplr). Non-parametric Spearman correlations were used to assess relationship between neutralization and binding or neutralization and effector function data as well as between neutralization data obtained from the pseudotyped and authentic SARS-CoV-2 neutralization assays. Two-tailed Mann–Whitney t-tests were used to verify the existence of significant differences between NTD and RBD mAbs in several binding and functional assays. In the animal studies, one-way ANOVA with Dunnett's multiple comparisons tests were used to assess significance in weight changes and viral loads across groups compared to the isotype control antibody-treated animals. Survival curves were compared individually to the isotype control antibody using a Mantel-Cox log-rank test. Fold change in binding to mutant proteins was calculated relative to the WA-1 Spike or RBD proteins. In absence of binding, a background binding value (0.05 nm in BLI assays) was attributed. Fold change in neutralization to VoC was calculated relative to the IL1/2020 virus. Non-neutralizing mAbs were assigned the $IC_{50}$ of 25 µg ml$^{-1}$ antibody, the mAb starting concentration in the assay. All tests, except for the 5-parameter logistic regression performed in R (version 3.6.3) and R studio (1.2.1355), were performed in Prism (version 9, GraphPad Software). Data were graphed using Prism software (version 9, GraphPad Software).

### Reporting summary

Further information on research design is available in the Nature Portfolio Reporting Summary linked to this article.

## Data availability

All other data are available in the main manuscript, Supplementary Information, or the Source Data file provided with this paper. Items described in this study will be made available to the scientific community by contacting M.G.J. or S.J.K. and upon completion of a materials transfer agreement. The associated accession numbers, coordinates and structure factors of the crystallographic complexes reported in this paper are available from the Protein Data Bank (PDB) with accession codes PDB: 8FI9 and 8FHY. The antibody sequences are available at Genbank with accession numbers OR078588-OR078627 and are provided in Supplementary Table 6. Antibody variable regions were synthesized and cloned (Genscript) into CMVR expression vectors (NIH AIDS reagent program) between a murine Ig leader (GenBank DQ407610) and the constant regions of human IgG1 (GenBank AAA02914), Igκ (GenBank AKL91145) or Igλ (GenBank AAA02915). Recombinant SARS-CoV-2 proteins RBD (318-514), NTD (1-290) and S1 (1-665) were made from a synthesized full-length Spike sequence (Genscript) from strain USA/IL1/2020 (GenBank # MN988713). Additional sequences used in this work include the coding sequence for SARS-CoV-2 (Genbank # MN908947), Wuhan-Hu-1 strain genome sequence (GenBank # MN9089473) stabilized trimer (S-2P), codon-optimized S expression plasmid encoding SARS-CoV-1 (Sino 1-11, GenBank # AY485277), and IL1/2020 isolate (GenBank # MN988713). Source data are provided with this paper.

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

## Acknowledgements

We thank M. Amare, M. Taddesse, E.Sondergaard, T. Mbehratu, J. Headley, J. Lay and S. Daye for programmatic, administrative, and regulatory support and planning. We also thank M. Creegan, and the MHRP Flow Core facility for help with FACS sorting, and E. Kavusak, S. Molnar, J. Heller, C. Kuklis, S. Soman, C. Kannadka, T. Lang, S. Muncil, K. Lanzer, T. Cookenham, F. Szaba, and M. Rao for technical support. This work was funded by the U.S. Department of Defense, Defense Health Agency, through the CARES Act, (N.L.M, K.M., S.J.K.) and MIDRP Grant MI220230 (M.G.J.). Funding was executed through a cooperative agreement (W81XWH-18-2-0040) between the U.S. Department of Defense and the Henry M. Jackson Foundation for the Advancement of Military Medicine, Inc (S.V). Additional support was provided by the Intramural Research Program of the Vaccine Research Center, NIAID, NIH (N.J.S). In addition, this work was supported by NIH Grant R01AI50111 to P.D.B. and R01AI78788 to T.H. as well as NIH contract HHSN 75N93019C00073 to B.J.D. The X-ray crystallographic work is based upon research conducted at the Northeastern Collaborative Access Team beamlines, which are funded by the National Institute of General Medical Sciences from the National Institutes of Health (P30 GM124165). The Eiger 16M detector on 24-ID-E is funded by a NIH-ORIP HEI grant (S10OD021527). This research used resources of the Advanced Photon Source, a U.S. Department of Energy (DOE) Office of Science User Facility operated for the DOE Office of Science by Argonne National Laboratory under Contract No. DE-AC02-06CH11357. Research was conducted in compliance with the Animal Welfare Act and other federal statutes and regulations relating to animals and experiments involving animals and adheres to principles stated in the *Guide for the Care and Use of Laboratory Animals*, NRC Publication, 1996 edition. The research protocol was approved by the Institutional Animal Care and Use Committee of the Trudeau Institute, protocol 20-007. K18-hACE2 transgenic mice were obtained from Jackson Laboratories (Bar Harbor, ME). Mice were housed in the animal facility of the Trudeau Institute and cared for in accordance with local, state, federal, and institutional policies in a National Institutes of Health American Association for Accreditation of Laboratory Animal Care-accredited facility. Material has been reviewed by the Walter Reed Army Institute of Research. There is no objection to its presentation and/or publication. The opinions or assertions contained herein are the private views of the authors, and are not to be construed as official, or as reflecting true views of the Department of the Army or the Department of

Defense or Henry M. Jackson Foundation for the Advancement of Military Medicine, Inc.

## Author contributions

Conceptualization, N.L.M., M.G.J., K.M., S.J.K.; Investigation, R.S.S., K.G.L., J.L.J., V.D., L.M-R., H.B., L.W., S.V.M., M.Z., D.A.W., S.T., A.H., W.C.C., W-H.C., G.C.D., N.J., H.A.D.K., C.G.L., E.J.M., P.A.R., C.E.P., F.S., T.J.H., D.K.D., L.W.K., S.P.C., J.K.W., S.K., B.S., L.S., I.S., P.V.T., U.T., J.R.C., D.L.B., E.D., B.J.D., T.H. P.D.B., D-P-P., W.W.R., M.R., N.J.S., G.D.G., V.P., M.G.J., S.J.K. Data Curation, R.S.S., K.G.L, J.L.J., V.D., W.R., M.G.J., S.J.K. Writing – Original Draft, R.S.S., K.G.L., J.L.J., V.D., S.J.K., M.G.J.; Writing – Review & Editing, All authors; Visualization, R.S.S., K.G.L., J.L.J., V.D., D.P-P., W.R., S.J.K., M.G.J.; Supervision, B.J.D., T.H., W.R., P.D.B., N.D.C., S.V., N.J.S., G.D.G., V.P., N.L.M., K.M., S.J.K., and M.G.J;. Funding Acquisition, B.J.D., P.D.B., N.L.M., K.M., M.G.J. N.D.C., and S.V.

## Competing interests

J.K.W., S.K., E.D., and B.J.D. are employees of Integral Molecular, B.J.D. is a shareholder of Integral Molecular. W.H.C, A.H., P.V.T., J.L.J., K.M. and M.G.J. are named inventors on provisional patents describing SpFN molecules. A patent was filed containing the mAbs described in this publication for authors S.J.K., K.G.L, V.D. and M.G.J. M.G.J. is named as an inventor on international patent application WO/2018/081318 and U.S. patent 10,960,070 entitled "Prefusion coronavirus spike proteins and their use. The other authors declare no competing interests.

## Additional information

Rajeshwer S. Sankhala[1,2,12], Kerri G. Lal[1,2,12], Jaime L. Jensen [1,2], Vincent Dussupt[1,2,3], Letzibeth Mendez-Rivera[2,3], Hongjun Bai [1,2,3], Lindsay Wieczorek[2,3], Sandra V. Mayer [1,2], Michelle Zemil [2,3], Danielle A. Wagner [4], Samantha M. Townsley[2,3], Agnes Hajduczki[1,2], William C. Chang [1,2], Wei-Hung Chen [1,2], Gina C. Donofrio[2,3], Ningbo Jian[2,3], Hannah A. D. King[2,3], Cynthia G. Lorang[4], Elizabeth J. Martinez [1,2], Phyllis A. Rees[1,2], Caroline E. Peterson[1,2], Fabian Schmidt[5], Tricia J. Hart[6], Debra K. Duso[6], Lawrence W. Kummer[6], Sean P. Casey[6], Jazmean K. Williams[7], Shruthi Kannan[7], Bonnie M. Slike [2,3], Lauren Smith[2,3], Isabella Swafford[2,3], Paul V. Thomas[1,2], Ursula Tran[2,3], Jeffrey R. Currier[8], Diane L. Bolton [2,3], Edgar Davidson[7], Benjamin J. Doranz [7], Theodora Hatziioannou [5], Paul D. Bieniasz[5,9], Dominic Paquin-Proulx [2,3], William W. Reiley[6], Morgane Rolland [1,2,3], Nancy J. Sullivan[4], Sandhya Vasan [1,2,3], Natalie D. Collins[1,8], Kayvon Modjarrad[1,11], Gregory D. Gromowski [8], Victoria R. Polonis[3], Nelson L. Michael[10], Shelly J. Krebs [1,3,13] ✉ & M. Gordon Joyce [1,2,13] ✉

¹Emerging Infectious Diseases Branch, Walter Reed Army Institute of Research, Silver Spring, MD, USA. ²Henry M. Jackson Foundation for the Advancement of Military Medicine, Bethesda, MD, USA. ³U.S. Military HIV Research Program, Walter Reed Army Institute of Research, Silver Spring, MD, USA. ⁴Vaccine Research Center, National Institute of Allergy and Infectious Diseases, National Institutes of Health, Bethesda, MD, USA. ⁵Laboratory of Retrovirology, The Rockefeller University, New York, NY, USA. ⁶Trudeau Institute, Saranac Lake, NY, USA. ⁷Integral Molecular, Philadelphia, PA, USA. ⁸Viral Diseases Branch, Walter Reed Army Institute of Research, Silver Spring, MD, USA. ⁹Howard Hughes Medical Institute, The Rockefeller University, New York, NY, USA. ¹⁰Center for Infectious Disease Research, Walter Reed Army Institute of Research, Silver Spring, MD, USA. ¹¹Present address: Vaccine Research and Development, Pfizer, Pearl River, New York, NY, USA. ¹²These authors contributed equally: Rajeshwer S. Sankhala, Kerri G. Lal. ¹³These authors jointly supervised this work: Shelly J. Krebs, M. Gordon Joyce. ✉e-mail: skrebs@hivresearch.org; gjoyce@eidresearch.org

