## [Peer Review File · Nature Communications]

Diverse array of neutralizing antibodies elicited upon Spike Ferritin Nanoparticle vaccination in rhesus macaquesReviewers' Comments:

Reviewer #1:

Remarks to the Author:

Sankhala et al., characterize the antibody response to SpFN adjuvanted with ALFQ in non-human primates. Through B-cell isolation and functional screening, 20 mAbs targeting RBD, NTD or the S2 Domain were identified, from which six neutralizing antibodies displayed broad sarbecovirus cross-reactivity. Two RBD mAbs WRAIR-5001 and WRAIR-5021 with distinct epitope specificity were selected for structural determination and protection assessment in vivo. The findings of this report would be informative for the efficacy of SpFN in human beings and for the rational design of pan-COV vaccines against coronaviruses. Notably, B cells were isolated from only one sample (NHP.02) and the number of mAbs was relatively limited. Therefore, the authors should be more careful in the data analysis, interpretation and conclusions. Some detailed comments are as follows.

Major:

1. Supplementary Fig. 1a, b. Please specify the time point when the plasma or serum was obtained. "The magnitude of serum binding antibodies from SpFN-vaccinated macaques were comparable to the magnitude of serum binding antibodies elicited by macaques vaccinated with two doses of mRNA-1273 (Supplementary Fig. 1a).", are there any comparisons of plasma or serum neutralizing activity between the two vaccines?
2. The sorting strategy is novel but multi-step, which might lead to the loss of a large number of cells. Generally, SpFN-targeting B cells also recognize NTD, RBD or S2, why are the NTD and RBD included in the sorting strategy? Please make sure whether CD20-AF700 was used as it was not mentioned in the gating strategy.
3. Supplementary Fig. 3a-b. Were 25 mAbs tested in an enzyme immunoassay (EIA)? Please show them all. If not, please explain the reason.
4. Neutralization and hACE2-blocking results of two S2 targeting mAbs (WRAIR-5015, WRAIR-5016) are not shown. Are they unable to neutralize the PSV or not tested?
5. Fig. 2g. Please show the KD values.
6. Fig. 3a and Supplementary Fig. 1b should be combined. To determine if mAb epitopes recapitulated the plasma polyclonal antibody targets, it is more rigorous to use the same panel of competitive mAbs (Fig. 2a, b, Fig. 3a, Supplementary Fig. 1b).
7. Please discuss the relationship between the preference of effector function and region specificity (lines 165-171).
8. "While WRAIR-5001 was able to bind with high affinity to SARS-CoV-2 Omicron subvariants BA.1 and BA.4/5 (Fig. 2g), these mAbs had diminished neutralization activity", can the authors explain the reason from the structural characteristic?

Minor:

1. Please clarify plasma or serum was used (line110), which is inconsistent with the figure legend.
2. "µg" is wrongly displayed in several figures.
3. Fig 1f legend, "the reciprocal IC50" should be "the IC50"; Line 1013, "10 mg ml⁻¹" should be 10 mg/kg; Line 1014, "Delta B.1.618.2" should be Delta B.1.617.2; other errors, lines 176, 191, 249, 394, 396, 402-404, 428.

Reviewer #2:

Remarks to the Author:

The manuscript describes the study of neutralizing antibodies elicited by a SARS-CoV-2 Spike ferritin nanoparticle (SpFN) vaccine in non-human primates, which yielded six neutralizing monoclonal antibodies that specifically targeted the Spike RBD, NTD or S2 domain and exhibited broad cross-reactivity against sarbecoviruses. The RBD-targeting monoclonal antibodies were classified into two major epitope categories; Group A RBD mAbs targeted the hACE2-binding site while Group B RBD

mAbs targeted a conserved cryptic site. WRAIR-5001, a mAb that recognized the cryptic epitope, provided complete protection against the Delta variant in a murine challenge study and exhibited high affinity binding to the conserved epitope, including the Omicron variants.

While this is a comprehensive study on neutralizing Abs isolation and characterization in NHPs, the overall conception and the antibodies yielded lacks significant novelty.

1. The greatest weakness of this paper is that the selected mAbs is not broadly neutralizing enough to cover Omicron variants or SARS-CoV-1. I assume that is the reason why the in vivo experiments were conducted only using one strain SARS-CoV-2 Delta (B.1.617.2).
2. Although structural studies of the RBD mAbs were performed, the reviewer's concern is that only the two mAbs cannot represent the epitopes targeted by SpFN-vaccinated macaques, for example, the NTD or S2 mAbs were not considered.
3. Fig.6b and Supplementary Fig. 1b, there is no color in the legend of the heatmaps.
4. In Supplementary Fig. 1c, I did not see the PBS line (gray line) mentioned in the figure legend. The X and Y axes do not align perfectly (same with Supplementary Fig. 1d).
5. Paying attention to details in formatting would enhance the manuscript. There were garbled unit symbols and misalignment in several figures, and some sentences contained double periods. For example, in Fig. 1h, the figure panels are not aligned; in Fig. 2c, the font size of the caption varied, etc.

Reviewer #3:

Remarks to the Author:

Sankhala and co-authors report a comprehensive analysis of the antibodies (Abs) raised by immunization of NHPs with SpFN vaccine and identification of the Spike (S) epitopes targeted by the Abs. This work is a continuation of their previous studies related to the design of the SpFN vaccine, a ferritin nanoparticle carrying 8 copies of trimeric stabilized Spikes, which is now in phase I clinical trials, and immunogenicity evaluations performed in mice, hamsters, and macaques (references 23-27 in the manuscript). Here they apply reverse vaccinology approach to identify the S epitopes, providing important findings such as that 1) the anti-RBD Abs bind to epitopes blocking the RBD interactions with the cellular receptor ACE-2 (so called 'Group A' Abs) or they recognize a cryptic epitope, away from the receptor binding site, which is conserved in sarbecoviruses ('Group B' Abs), 2) Ab WRAIR-5021 from Group A provides full protection from Delta strain in murine challenge experiment (Group B Ab did not protect). These results are noteworthy and will serve as a guide for development and optimization of vaccines and prophylactic in vivo strategies. I would like to praise the authors for the very well-structured and clear manuscript.

Major comments:

1. The manuscript would benefit from an explicit description of what is novel regarding the epitopes and Abs described here. The WRAIR-5001 Ab recognizes a cryptic epitope that is well-conserved, which is very interesting. Is this the first Ab of this type? The authors mention S309 Ab (line 335), but don't elaborate how similar or different WRAIR-5001 is to S309. Also, was there anything unusual, or not previously seen in the polyclonal response? Is there something interesting in site G that has weak binding for mRNA-vaccine-derived plasma (lines 113-115)? A novel sorting strategy is mentioned in line 91 – could the authors explain in the text what was new?
2. Lines 244-246: Is there a structural explanation (that could be derived from the presented structures) for potent neutralization of Delta variant by WRAIR-5021 and weaker neutralization by WRAIR-5001 Ab?
3. The structure of RBD bound to WRAIR-5001 Fab was determined at, for X-ray crystallography, low resolution of 4.3Å. Could the authors comment on the quality of electron density maps in the epitope-CDR regions, and if the side chains could be ambiguously placed to allow the type of analysis presented in lines 301-331.
4. Lines 402-404: Ab WRAIR-5021 binds to a series of variants, including Omicron BA4.5, but potent

neutralization is not observed for most of Omicron variants. Could the authors speculate on the reasons behind, and is there anything in the structure of RBD bound to WRAIR-5021 Fab that could explain why neutralization might be poorer than the binding would suggest?

Minor comments:

5. Was there a special reason to use tetramerized subdomains of NTD and RBD (line 134) since the S forms a trimer, and trimeric constructs would maybe make more sense?

6. Lines 154-156: Could the authors clarify what they mean by this sentence? I am confused with what cross-binding refers to and what is being compared. What could be the reasons that 2 mAbs are hexaPro-specific?

7. Line 221 – consider replacing 'targeted towards' with 'located in'.

8. Lines 427-430: For the sake of readers who are not immunologists, explanation of what roles ADCP and ADCC play in immune response, and how this is relevant for vaccine design would be helpful.

9. Consider including a drawing of a model showing trimeric S and two Fabs (5001 and 5021) and how different exposure of epitopes may explain some of the neutralization / binding data / in vivo data.

10. Table S3: The resolution cut-off for 8F19 seems a bit generous, considering the 200% Rmerge, $I/\sigma I$ of 0.3 and $CC(1/2)$ of 35% for the highest resolution shelf. It would help to include in SI some images that show electron density in the regions where epitopes and CDRs interact.

11. Table S3: The $I/\sigma I$ for 8FHY is 1.6 for the highest resolution shelf, while the table on page 11 of the PDB report for 8FHY indicates value of 0.25. Could the authors comment on this?

Response to Reviewers

We thank all reviewers for their thorough reviews, questions, and suggestions. We have significantly revised this manuscript, specifically editing Fig. 2 and Fig. 3 and adding Supplementary Figures 5-8 to improve clarity. We hope that these revisions will satisfy the reviewer comments.

Reviewer #1 (Remarks to the Authors):

Sankhala et al., characterize the antibody response to SpFN adjuvanted with ALFQ in non-human primates. Through B-cell isolation and functional screening, 20 mAbs targeting RBD, NTD or the S2 Domain were identified, from which six neutralizing antibodies displayed broad sarbecovirus cross-reactivity. Two RBD mAbs WRAIR-5001 and WRAIR-5021 with distinct epitope specificity were selected for structural determination and protection assessment in vivo. The findings of this report would be informative for the efficacy of SpFN in human beings and for the rational design of pan-COV vaccines against coronaviruses. Notably, B cells were isolated from only one sample (NHP.02) and the number of mAbs was relatively limited. Therefore, the authors should be more careful in the data analysis, interpretation and conclusions. Some detailed comments are as follows.

Response:

We thank Reviewer #1 for their supportive and constructive comments and suggestions. We have taken their comments into account and made modifications throughout the manuscript accordingly and specifically as outlined below. Notably, we made a comment in the Discussion about the limitations of this study, including the number of mAbs isolated from one animal. We appreciate the recognition of the importance of evaluating antibody specificity in NHPs with our SpFN vaccine platform as part of a larger effort to develop protective pan-CoV vaccines.

Major comments:

1. Supplementary Fig. 1a, b. Please specify the time point when the plasma or serum was obtained. “The magnitude of serum binding antibodies from SpFN-vaccinated macaques were comparable to the magnitude of serum binding antibodies elicited by macaques vaccinated with two doses of mRNA-1273 (Supplementary Fig. 1a).”, are there any comparisons of plasma or serum neutralizing activity between the two vaccines?

Response:

We sincerely appreciate this comment, as comparing these vaccine responses helps to put these data into context. The timepoints used were 2 weeks after the 2nd vaccination for both mRNA-1273- and SpFN-vaccinated rhesus macaques. Neutralizing titers were previously published for both mRNA-1273^{1,2} and SpFN³ vaccinated rhesus macaques, as referenced, and were found to be comparable at the 2 weeks post boost time point for both vaccine strategies. Taking the lentivirus pseudotyped virus neutralization assays that were performed for both studies at this same timepoint, median ID50 neutralization titers were found as follows: mRNA-1273 vaccination: D614G: 4,700; Epsilon, 5,900; Gamma, 1,500; Lambda: 1,100, Delta: 830, and

Beta: 770. SpFN vaccination: WA-1: 52,723, Alpha: 25,003, and Beta: 10,209. Clarifying statements were added to the Results on the timepoints used and the referenced neutralization data to provide context.

2. The sorting strategy is novel but multi-step, which might lead to the loss of a large number of cells. Generally, SpFN-targeting B cells also recognize NTD, RBD or S2, why are the NTD and RBD included in the sorting strategy? Please make sure whether CD20-AF700 was used as it was not mentioned in the gating strategy.

Response:

With any sorting strategy into plates, we agree that there is cell loss. However, we have optimized this sequential sorting technique using the SpFN probes to minimize cell loss, comparable to levels if using probes only without sequential staining. In our previous work, we used SpFN, as well as RBD and NTD, to successfully isolate mAbs from convalescent donors. These studies revealed B cells binding to SpFN, but also exclusively to the tetramerized NTD and RBD antigens, hence their inclusion in the panel⁴. As the reviewer indicates, previous QC experiments demonstrated competition between the RBD and NTD antigenic tetramers and the SpFN molecule. Interestingly this level of competition did not make a difference when isolating B cells from convalescent donors. In this sorting strategy, we added SpFN¹, displaying the S from SARS-CoV-1, along with SpFN, displaying the S on SARS-CoV-2. As shown in Supplementary Fig. 2c, sequential staining using SpFN and SpFN¹ bound the majority of the SARS-specific B cells, leaving only a minority to bind to the RBD or NTD tetramers. We have added statements within the Results to explain how we used this sorting strategy to prioritize B cell binding to the SpFN molecules. In addition, CD20-AF700 was included in the panel table in error and has been removed.

3. Supplementary Fig. 3a-b. Were 25 mAbs tested in an enzyme immunoassay (EIA)? Please show them all. If not, please explain the reason.

Response:

Supplementary Fig. 3a-b have been updated to display all 25 mAbs. Only the SpFN-reactive mAbs were tested for ferritin particle binding, and thus, Supplementary Fig. 3c remains the same.

4. Neutralization and hACE2-blocking results of two S2 targeting mAbs (WRAIR-5015, WRAIR-5016) are not shown. Are they unable to neutralize the PSV or not tested?

Response:

WRAIR-5015 and WRAIR-5016 had no detectable neutralization against WA-1, so were not displayed in the Figure 1. Their neutralization curves have been added as Supplementary Fig. 4d to satisfy the reviewer's request. Each were also tested for ACE2 inhibition with minimal blocking measured, as shown now in Supplementary Fig. 5c.

5. Fig. 2g. Please show the KD values.

Response:

Thank you. We have revised this figure to include the K_D values. This data is now displayed in Figure 3e.

6. Fig. 3a and Supplementary Fig. 1b should be combined. To determine if mAb epitopes recapitulated the plasma polyclonal antibody targets, it is more rigorous to use the same panel of competitive mAbs (Fig. 2a, b, Fig. 3a, Supplementary Fig. 1b).

Response:

The reviewer brings to light a good point that mapping the mAbs in a similar manner to the serum would make for a more cohesive summary of the target of SpFN vaccination. In response, we tested each of the mAbs outlined in the serum competitions (Supplementary Fig. 1b) with the WRAIR NTD- and RBD-targeting mAbs to compare the serum-specificity with the mAb specificity. We found that the WRAIR NTD- and RBD- mAbs competed with mAbs identified in serum mapping, with the notable RBD class III S309 target recapitulated by our WRAIR Group B mAbs. We created new Supplementary Figs. 5-6 to better demonstrate the overlap in specificity found between the serum and the mAbs, and modified to Results to add clarity on these comparisons.

7. Please discuss the relationship between the preference of effector function and region specificity (lines 165-171).

Thank you for this comment. Antibody domain specificity and Fc effector functions, beyond that of neutralization, is currently understudied. In response to this comment, we have included more background on Fc effector functions in the context of SARS-CoV-2 and have added text to include comparisons that our group has made previously from mAbs isolated from convalescent donors, with additional literature citations of effector functions in the context of both SARS-CoV-2 vaccination and natural infection.

8. “While WRAIR-5001 was able to bind with high affinity to SARS-CoV-2 Omicron subvariants BA.1 and BA.4/5 (Fig. 2g), these mAbs had diminished neutralization activity”, can the authors explain the reason from the structural characteristic?

Response:

As has been observed for multiple RBD-directed antibodies and previously described by several groups^{5, 6, 7, 8, 9}, the Omicron subvariants are highly resistant to antibody-mediated neutralization. This is due to the structural dynamics of the Omicron spike protein, as Omicron-specific mutations allow for both immune escape and greater affinity for ACE2. None of the VoC mutations (including those found in the Omicron subvariants) are found at the WRAIR-5001 epitope, which corresponds to the class V RBD-directed antibody group. As WRAIR-5001 binds the highly-conserved, cryptic epitope that is distal to the ACE2 binding site, antibody binding to spike (either WA-1 or VoC) poorly competes with ACE2 binding (Fig. 3a-b). Thus, binding of WRAIR-5001 to the RBD is unaffected by Omicron mutations, while these mutations stabilize the spike and allow for increased affinity for ACE2, likely explaining the diminished neutralization potency. Statements have been added to the Results and Discussion to further clarify these points.

Minor comments:

1. Please clarify plasma or serum was used (line110), which is inconsistent with the figure legend.

Response:

The sampling was all from serum, not plasma, and these errors have been corrected throughout the manuscript including its mention in line110. We thank the reviewer for highlighting this.

2. “ μg ” is wrongly displayed in several figures.

Response:

We have closely scrutinized the manuscript and figures and have corrected these formatting issues.

3. Fig 1f legend, “the reciprocal IC50” should be “the IC50”; Line 1013, “10 mg ml⁻¹” should be 10 mg/kg; Line 1014, “Delta B.1.618.2” should be Delta B.1.617.2; other errors, lines 176, 191, 249, 394, 396, 402-404, 428.

Response:

We thank Reviewer #1 for calling our attention to these typos and have corrected them throughout the manuscript.

Reviewer #2 (Remarks to the Authors):

The manuscript describes the study of neutralizing antibodies elicited by a SARS-CoV-2 Spike ferritin nanoparticle (SpFN) vaccine in non-human primates, which yielded six neutralizing monoclonal antibodies that specifically targeted the Spike RBD, NTD or S2 domain and exhibited broad cross-reactivity against sarbecoviruses. The RBD-targeting monoclonal antibodies were classified into two major epitope categories; Group A RBD mAbs targeted the hACE2-binding site while Group B RBD mAbs targeted a conserved cryptic site. WRAIR-5001, a mAb that recognized the cryptic epitope, provided complete protection against the Delta variant in a murine challenge study and exhibited high affinity binding to the conserved epitope, including the Omicron variants.

While this is a comprehensive study on neutralizing Abs isolation and characterization in NHPs, the overall conception and the antibodies yielded lacks significant novelty.

Response:

We thank Reviewer #2 for their constructive comments and suggestions. We have taken their comments into account and made modifications throughout the manuscript specifically as outlined below. Notably, while we agree that the epitopes of the mAbs described do overlap with previously described classes, the utilization of ferritin nanoparticles, such as SpFN, as a vaccine strategy is currently understudied. Exploring the targets of neutralizing responses is important to give insight into the observed breadth of immune response and neutralization, correlates of protection, and inform the improvement of future vaccine strategies. We have added text in the manuscript (Introduction and Discussion) to emphasize these points.

Major comments:

1. The greatest weakness of this paper is that the selected mAbs is not broadly neutralizing enough to cover Omicron variants or SARS-CoV-1. I assume that is the reason why the *in vivo* experiments were conducted only using one strain SARS-CoV-2 Delta (B.1.617.2).

Response:

While the reviewer is correct that WRAIR Group A mAbs, such as WRAIR-5021, do not neutralize Omicron subvariants, WRAIR Group B mAbs, such as WRAIR-5001, do demonstrate modest neutralization of Omicrons BA.1, BA.2, and BA.5, as well as SARS-CoV-1 (Fig. 3d and Fig. 1g). These mAbs give insight into the target of cross-sarbecovirus neutralization observed in the IgG fraction of the plasma in macaques. We have now re-emphasized these points in the text in several places to highlight the cross-sarbecovirus neutralization potential of WRAIR-5001 and the WRAIR RBD Group B mAbs. As these mAbs were isolated after 2 vaccinations of SpFN, we hypothesize that Omicron neutralization may expand after a 3rd SpFN vaccination. Indeed, results from a Phase I clinical trial using SpFN/ALFQ have yielded serum neutralization of Omicron subvariants, including XBB.1.5 after three SpFN/ALFQ vaccinations (*pending acceptance, Lancet Microbe*).

The Delta variant was chosen as the challenge strain of SARS-CoV-2 in our *in vivo* studies since it was the circulating strain at the time of experimentation and recommended VoC of relevance.

We did not use the WA-1 strain, since we anticipated that both Group A WRAIR-5021 and Group B WRAIR-5001 would protect 100% against a WA-1 viral challenge, based on their neutralization titers. Neutralization to WA-1 was previously shown to correlate with *in vivo* protection for RBD mAbs⁴. However, against Delta, RBD-A WRAIR-5021 demonstrated potent neutralization (IC₅₀=0.002 µg/ml), whereas RBD-B WRAIR-5001 yielded more modest neutralization (IC₅₀=1.5 µg/ml, Fig. 3d), and we wanted to determine if neutralization potency also correlated with *in vivo* protection against Delta challenge. We thank the reviewer for these comments, and we have revised the text in the Results to emphasize these points.

2. Although structural studies of the RBD mAbs were performed, the reviewer's concern is that only the two mAbs cannot represent the epitopes targeted by SpFN-vaccinated macaques, for example, the NTD or S2 mAbs were not considered.

Response:

We thank the reviewer for this comment and agree with this comment; the number of mAbs isolated in this study was limited and does not fully represent all of the epitopes targeted by SpFN-vaccinated macaques. However, even with the limited mAbs isolated, these mAbs identified recapitulate the antibody responses found in the serum and represent the cross-neutralization and functional protection found in the macaques. We have modified statements within the Discussion to reemphasize these points.

In addition, our group put forth significant effort since receiving the manuscript review to perform structural studies on the the NTD- or S2 mAbs. Despite our best efforts, we were unable to add additional data, even at the level of low-resolution, negative stain electron microscopy (nsEM). We worked with S-2P and HexaPro constructs, and Fabs of NTD-directed Abs WRAIR-5009, WRAIR-5010, and WRAIR-5013, as well as S2-targeted Abs WRAIR-5015, WRAIR-5016, and WRAIR-5018. However, we were unable to collect suitable structural data to include in this manuscript. We also included these limitations within the Discussion.

3. Fig.6b and Supplementary Fig. 1b, there is no color in the legend of the heatmaps.

Response:

We apologize for this oversight and have updated our figures to include the heatmap scales.

4. In Supplementary Fig. 1c, I did not see the PBS line (gray line) mentioned in the figure legend. The X and Y axes do not align perfectly (same with Supplementary Fig. 1d).

Response:

We thank the reviewer for their attention to detail and have amended the Figures appropriately. Likewise, we have reviewed all figures to ensure better formatting.

5. Paying attention to details in formatting would enhance the manuscript. There were garbled unit symbols and misalignment in several figures, and some sentences contained double periods. For example, in Fig. 1h, the figure panels are not aligned; in Fig. 2c, the font size of the caption varied, etc.

Response:

We thank the reviewer for pointing out these errors, and edited the manuscript to reformat the figures correctly, including aligning the subfigures and paying attention to font size and symbols. We have corrected these formatting issues throughout the figures and manuscript.

Reviewer #3 (Remarks to the Authors):

Sankhala and co-authors report a comprehensive analysis of the antibodies (Abs) raised by immunization of NHPs with SpFN vaccine and identification of the Spike (S) epitopes targeted by the Abs. This work is a continuation of their previous studies related to the design of the SpFN vaccine, a ferritin nanoparticle carrying 8 copies of trimeric stabilized Spikes, which is now in phase I clinical trials, and immunogenicity evaluations performed in mice, hamsters, and macaques (references 23-27 in the manuscript). Here they apply reverse vaccinology approach to identify the S epitopes, providing important findings such as that 1) the anti-RBD Abs bind to epitopes blocking the RBD interactions with the cellular receptor ACE-2 (so called ‘Group A’ Abs) or they recognize a cryptic epitope, away from the receptor binding site, which is conserved in sarbecoviruses (‘Group B’ Abs), 2) Ab WRAIR-5021 from Group A provides full protection from Delta strain in murine challenge experiment (Group B Ab did not protect). These results are noteworthy and will serve as a guide for development and optimization of vaccines and prophylactic in vivo strategies. I would like to praise the authors for the very well-structured and clear manuscript.

Response:

We thank Reviewer #3 for their kind commendation and recommendations to improve the clarity of the manuscript and figures.

Major comments:

1. The manuscript would benefit from an explicit description of what is novel regarding the epitopes and Abs described here. The WRAIR-5001 Ab recognizes a cryptic epitope that is well-conserved, which is very interesting. Is this the first Ab of this type? The authors mention S309 Ab (line 335), but don’t elaborate how similar or different WRAIR-5001 is to S309. Also, was there anything unusual, or not previously seen in the polyclonal response? Is there something interesting in site G that has weak binding for mRNA-vaccine-derived plasma (lines 113-115)? A novel sorting strategy is mentioned in line 91 – could the authors explain in the text what was new?

Response:

We thank the reviewer for this comment. The epitopes of the mAbs described do overlap with previously described classes. However, the utilization of ferritin nanoparticles, such as SpFN, as a vaccine strategy is currently understudied and exploring the targets of neutralizing responses is important to give insight into the broad response observed with this vaccine class, correlates of protection, and informing the improvement of future vaccine strategies. We classify WRAIR-5001 as a class III/V RBD-directed Ab based upon its epitope; it is not the first Ab of this type, nor the first of the class V Abs characterized by our group^{4,10}. We included figures outlining the overlap in epitopes between WRAIR-5001 and S309 (Fig. 5g) and have enhanced this description in the manuscript text. In addition, we have amended the text to describe how the novelty of the immune response after SpFN vaccination differs compared to that of the response elicited after mRNA vaccination (which is the dominant mode of SARS-COV-2 vaccination in use at the moment), which uniquely target antigenic sites A, C and D on the NTD (A and D are commonly neutralizing) and site I on the RBD, which is also a neutralizing epitope. Finally, we

added additional text clarifying why our sorting strategy was novel, in which we prioritized the SpFN vaccine molecules themselves as SARS-CoV BCR bait.

2. Lines 244-246: Is there a structural explanation (that could be derived from the presented structures) for potent neutralization of Delta variant by WRAIR-5021 and weaker neutralization by WRAIR-5001 Ab?

Response:

The WRAIR-5021 epitope overlaps extensively with that of ACE2 (Fig. 4b) and binding by this Ab completely inhibits ACE2 binding to WA-1 spike and isolated RBD. There is only one mutation in the Delta variant found at the WRAIR-5021 epitope: T478K. The flexibility of the lysine butylammonium side chain likely does not interfere with Ab binding and may form a salt bridge with S93 or D94 of the WRAIR-5021 light chain. We have included a Supplementary Fig. 7 to aid the reader in the visualization of this structural explanation. Weaker neutralization by WRAIR-5001 for all VOCs, not just Delta, may be explained by the stabilization of the spike trimer by these mutations and the resultant increase in RBD affinity for ACE2.

3. The structure of RBD bound to WRAIR-5001 Fab was determined at, for X-ray crystallography, low resolution of 4.3Å. Could the authors comment on the quality of electron density maps in the epitope-CDR regions, and if the side chains could be ambiguously placed to allow the type of analysis presented in lines 301-331.

Response:

As the Reviewer notes, the structures of the RBD with WRAIR-5001 Fab was determined at a relatively low resolution. However, the electron density was of sufficient quality to allow for unambiguous placement of the RBD and heavy and light Fab chains during molecular replacement. To highlight this, we have included figures showing the 2Fo-Fc map along the epitope-CDR regions, as requested by the Reviewer (Supplementary Fig. 8).

4. Lines 402-404: Ab WRAIR-5021 binds to a series of variants, including Omicron BA4.5, but potent neutralization is not observed for most of Omicron variants. Could the authors speculate on the reasons behind, and is there anything in the structure of RBD bound to WRAIR-5021 Fab that could explain why neutralization might be poorer than the binding would suggest?

Response:

As has been observed for multiple RBD-directed antibodies and previously described by several groups^{5, 6, 7, 8, 9}, the Omicron subvariants are highly resistant to antibody-mediated neutralization. This is due to the structural dynamics of the Omicron spike protein, as Omicron-specific mutations allow for both immune escape and greater affinity for ACE2. None of the VoC mutations (including those found in the Omicron subvariants) are found at the WRAIR-5001 epitope, which corresponds to the class V RBD-directed antibody group. As WRAIR-5001 binds the highly-conserved, cryptic epitope that is distal to the ACE2 binding site, antibody binding to spike (either WA-1 or VoC) poorly competes with ACE2 binding (Fig. 3a-b). Thus, binding of WRAIR-5001 to the RBD is unaffected by Omicron mutations, while these Omicron spike mutations stabilize the closed spike and reduce the time that the WRAIR-5021 epitope is available and this likely explains the diminished neutralization potency.

Minor comments:

5. Was there a special reason to use tetramerized subdomains of NTD and RBD (line 134) since the S forms a trimer, and trimeric constructs would maybe make more sense?

Similar to the comment by Reviewer 1, we used SpFN, as well as RBD and NTD, to successfully isolate mAbs from convalescent donors. These studies revealed B cells binding to SpFN, but also exclusively to the tetramerized NTD and RBD antigens, hence their inclusion in the panel⁴. In the current sorting strategy, we added SpFN¹, displaying the trimeric S on SARS-CoV-1, along with SpFN, displaying the trimeric S on SARS-CoV-2. As shown in Supplementary Fig. 2c, sequential staining using SpFN and SpFN¹ bound the majority of the SARS-specific B cells, leaving only a minority to bind to the RBD or NTD tetramers. We have added statements within the Results to explain how we used this sorting strategy to prioritize B cell binding to the SpFN molecules.

6. Lines 154-156: Could the authors clarify what they mean by this sentence? I am confused with what cross-binding refers to and what is being compared. What could be the reasons that 2 mAbs are hexaPro-specific?

The text has been amended to state these mAbs “also bind,” as opposed to “cross-bind,” which we acknowledge is a misleading statement. We hypothesize that the mAbs that can bind both Hexapro (S-protein) and their respective S protein subdomain recognize an epitope that is found in both the tertiary (isolated subdomain) and quaternary (S-protein trimer) structures. However, the mAbs that only bind the S-protein trimer only recognize quaternary epitopes not found on the isolated subdomains. We have also added text here to clarify this point.

7. Line 221 – consider replacing ‘targeted towards’ with ‘located in’.

Response:

Thank you. We have made the modification as indicated.

8. Lines 427-430: For the sake of readers who are not immunologists, explanation of what roles ADCP and ADCC play in immune response, and how this is relevant for vaccine design would be helpful.

This is a point well taken, and we have added text and several citations to the Results and Discussion of the manuscript to describe not only what these functions are associated with in the context of SARS-CoV-2 vaccination or natural infection, but also their general viral clearance and the immune cells most frequently attributed to these functions.

9. Consider including a drawing of a model showing trimeric S and two Fabs (5001 and 5021) and how different exposure of epitopes may explain some of the neutralization / binding data / in vivo data.

Response:

We thank Reviewer #3 for this suggestion and have incorporated this model into Supplementary Fig. 7.

10. Table S3: The resolution cut-off for 8FI9 seems a bit generous, considering the 200% Rmerge, $I/\sigma I$ of 0.3 and CC(1/2) of 35% for the highest resolution shelf. It would help to include in SI some images that show electron density in the regions where epitopes and CDRs interact.

Response:

We have updated the data collection statistics with values from additional resolution shells (Supplementary Table 3) and have included figures of the RBD with WRAIR-5001 (8FI9) with the electron density at the RBD-Fab interface shown contoured at 1σ (Supplementary Fig. 8).

11. Table S3: The $I/\sigma I$ for 8FHY is 1.6 for the highest resolution shelf, while the table on page 11 of the PDB report for 8FHY indicates value of 0.25. Could the authors comment on this?

Response:

As is common practice, we deposited the structure factors from the final round of refinement, although there is renewed effort within the structural biology community to require deposition of raw data. The $I/\sigma I$ reported in Supplementary Table 3 is the $I/\sigma I$ observed after data processing, prior to refinement, and the $I/\sigma I$ displayed in the deposition report was determined post-refinement. This difference is, understandably, notable, as very few (41) unique reflections were automatically discarded during refinement. We performed further refinement in REFMAC5, using the same data and similar refinement parameters, and found that, although we obtained nearly identical refinement statistics to the Phenix output, the PDB validation server reported the $I/\sigma I$ from REFMAC5 as ~ 1.0 .

To verify these values, we consulted the PDB curation team responsible for handling the deposition of this structure. The consensus was that the discrepancy is due to the statistics that the data processing and refinement programs report. HKL-2000 was used to process this dataset, and the shell statistics that HKL-2000 outputs after merging and scaling contain $\text{avg}(I)/\text{avg}(\sigma I)$ instead of $\text{avg}(I/\sigma I)$. For standard datasets, $\text{avg}(I)/\text{avg}(\sigma I)$ yields a larger number than $\text{avg}(I/\sigma I)$, particularly if the low resolution data are very strong and well-measured, or if the high resolution data are noisy.

References:

1. Corbett KS, *et al.* Protection against SARS-CoV-2 Beta variant in mRNA-1273 vaccine-boosted nonhuman primates. *Science* **374**, 1343-1353 (2021).
2. Gagne M, *et al.* Protection from SARS-CoV-2 Delta one year after mRNA-1273 vaccination in rhesus macaques coincides with anamnestic antibody response in the lung. *Cell* **185**, 113-130 e115 (2022).
3. Joyce MG, *et al.* A SARS-CoV-2 ferritin nanoparticle vaccine elicits protective immune responses in nonhuman primates. *Sci Transl Med* **14**, eabi5735 (2022).
4. Dussupt V, *et al.* Low-dose in vivo protection and neutralization across SARS-CoV-2 variants by monoclonal antibody combinations. *Nat Immunol* **22**, 1503-1514 (2021).
5. Dejnirattisai W, *et al.* SARS-CoV-2 Omicron-B.1.1.529 leads to widespread escape from neutralizing antibody responses. *Cell* **185**, 467-484 e415 (2022).
6. Guo H, *et al.* Structures of Omicron spike complexes and implications for neutralizing antibody development. *Cell Rep* **39**, 110770 (2022).
7. Kumar S, *et al.* Structural insights for neutralization of Omicron variants BA.1, BA.2, BA.4, and BA.5 by a broadly neutralizing SARS-CoV-2 antibody. *Sci Adv* **8**, eadd2032 (2022).
8. Mannar D, *et al.* SARS-CoV-2 Omicron variant: Antibody evasion and cryo-EM structure of spike protein-ACE2 complex. *Science* **375**, 760-764 (2022).
9. Hoffmann M, *et al.* The Omicron variant is highly resistant against antibody-mediated neutralization: Implications for control of the COVID-19 pandemic. *Cell* **185**, 447-456 e411 (2022).
10. Jensen JL, *et al.* Targeting the Spike Receptor Binding Domain Class V Cryptic Epitope by an Antibody with Pan-Sarbecovirus Activity. *J Virol* **97**, e0159622 (2023).

Reviewers' Comments:

Reviewer #1:

Remarks to the Author:

All questions were addressed. No further concern.

Reviewer #2:

Remarks to the Author:

The reviewer appreciates the effort of the authors providing the revised version. While the study's novel sorting strategy for isolating specific monoclonal antibodies from SpFN-vaccinated macaques holds promise, the weak neutralization observed with some mAbs, the authors made efforts to provide an explanation. However, the number of antibodies obtained in this study was limited. Consequently, the identified antibodies may not fully represent the entirety of epitopes targeted by the SpFN vaccine. Furthermore, the antibodies yielded in this study may not introduce entirely new epitopes, and their broad-spectrum neutralization effects against coronaviruses may be somewhat constrained. The evaluation of immune responses elicited by these monoclonal antibodies in in vivo experiments may be considered somewhat limited. The review is not convinced that the current version of this manuscript reaches the threshold of Nat Commun.

Reviewer #3:

Remarks to the Author:

All the comments were appropriately addressed. The manuscript and figures have been improved. I have no further remarks.

Response to Reviewers

We thank all reviewers for their thorough reviews, questions, and suggestions. We have revised the manuscript, guided by the editor's instructions.

REVIEWERS' COMMENTS

Reviewer #1 (Remarks to the Author):

All questions were addressed. No further concern.

Response

We appreciate the reviewer's positive response.

Reviewer #2 (Remarks to the Author):

The reviewer appreciates the effort of the authors providing the revised version. While the study's novel sorting strategy for isolating specific monoclonal antibodies from SpFN-vaccinated macaques holds promise, the weak neutralization observed with some mAbs, the authors made efforts to provide an explanation. However, the number of antibodies obtained in this study was limited. Consequently, the identified antibodies may not fully represent the entirety of epitopes targeted by the SpFN vaccine. Furthermore, the antibodies yielded in this study may not introduce entirely new epitopes, and their broad-spectrum neutralization effects against coronaviruses may be somewhat constrained. The evaluation of immune responses elicited by these monoclonal antibodies in in vivo experiments may be considered somewhat limited. The review is not convinced that the current version of this manuscript reaches the threshold of Nat Commun.

Response

We appreciate the reviewer's positive comments, and on balance, we believe the paper reaches the appropriate standard.

Reviewer #3 (Remarks to the Author):

All the comments were appropriately addressed. The manuscript and figures have been improved. I have no further remarks.

Response

We appreciate the reviewer's positive response.